# Self-Improvement for Neural Combinatorial Optimization: Sample Without Replacement, but Improvement

**Jonathan Pirnay**                                      *jonathan.pirnay@tum.de*
*Technical University of Munich, TUM Campus Straubing*
*University of Applied Sciences Weihenstephan-Triesdorf*

**Dominik G. Grimm**                          *dominik.grimm@{hswt,tum}.de*
*Technical University of Munich, TUM Campus Straubing*
*University of Applied Sciences Weihenstephan-Triesdorf*

**Reviewed on OpenReview:** *https://openreview.net/forum?id=agT8ojoHOX*

## Abstract

Current methods for end-to-end constructive Neural Combinatorial Optimization usually train a policy using behavior cloning from expert solutions or policy gradient methods from reinforcement learning. While behavior cloning is straightforward, it requires expensive expert solutions, and policy gradient methods are often computationally demanding and complex to fine-tune. In this work, we bridge the two and simplify the training process by sampling multiple solutions for random instances using the current model in each epoch and then selecting the best solution as an expert trajectory for supervised imitation learning. To achieve progressively improving solutions with minimal sampling, we introduce a method that combines round-wise Stochastic Beam Search with an update strategy derived from a provable policy improvement. This strategy refines the policy between rounds by utilizing the advantage of the sampled sequences with almost no computational overhead. We evaluate our approach on the Traveling Salesman Problem and the Capacitated Vehicle Routing Problem. The models trained with our method achieve comparable performance and generalization to those trained with expert data. Additionally, we apply our method to the Job Shop Scheduling Problem using a transformer-based architecture and outperform existing state-of-the-art methods by a wide margin.

## 1 Introduction

Combinatorial Optimization (CO) plays a critical role in many real-world applications in fields as diverse as logistics, manufacturing, genomics and synthetic biology (Sbihi & Eglese, 2010; Crama, 1997; Naseri & Koffas, 2020). In a CO problem, the goal is to find the best solution from a finite set of options that maximizes a given objective function. The NP-hard nature of these problems and their complex variations make them exceptionally difficult to solve. Traditional methods for solving CO problems typically rely on exact algorithms, heuristics, and metaheuristics based on decades of research. Exact algorithms such as enumeration, cutting plane, and branch-and-bound solve CO problems to optimality, but are limited by NP-hardness (Laporte, 1992). On the other hand, (meta-)heuristic algorithms such as local search, genetic algorithms, and ant colony optimization can provide solutions faster without guaranteed quality (Crama et al., 2005; Yang, 2010; Dorigo et al., 2006). While powerful, the traditional methods struggle with scalability, computational efficiency, and adaptability to other problems. In addition, designing algorithms for a CO problem requires expert intuition and significant domain knowledge (Vesselinova et al., 2020).

To overcome these limitations, the success of deep learning has led to the emergence of Neural Combinatorial Optimization (NCO), which departs from traditional CO methods to leverage the pattern recognition and generalization capabilities of neural networks (Bengio et al., 2021). NCO aims to build models that can

generate approximate solutions to CO problems by learning from data without manually crafting algorithmic rules.

A prominent paradigm in NCO is the *constructive* approach, where a solution to a problem is built step by step in a sequential decision-making process. A neural network models the policy that guides these incremental decisions. This policy network is typically trained either by supervised learning (SL) (Vinyals et al., 2015; Joshi et al., 2019; Fu et al., 2021; Kool et al., 2022; Hottung et al., 2020; Drakulic et al., 2023; Luo et al., 2023) or by reinforcement learning (RL) (Bello et al., 2016; Kool et al., 2019b; Nazari et al., 2018; Chen & Tian, 2019; Ahn et al., 2020; d O Costa et al., 2020; Hottung & Tierney, 2020; Kwon et al., 2020; Ma et al., 2021; Kim et al., 2021; Park et al., 2021; Wu et al., 2021; Park et al., 2022; Zhang et al., 2020; Hottung et al., 2022; Zhang et al., 2024). SL-based methods use expert solutions as labeled data for behavior cloning. While this training scheme is simple and effective, obtaining sufficient high-quality solutions from (exact) solvers can be expensive or even impossible for complex problems. On the other hand, RL methods exploit the natural modeling of the sequential problem as a finite Markov decision process, where the objective function is used as an episodic reward to maximize. In contrast to SL methods, they do not require pre-generated solutions but can suffer from strong hyperparameter sensitivity (Schulman et al., 2017; Henderson et al., 2018) and sparse rewards. State-of-the-art RL methods for NCO perform remarkably on the training distribution but generalize poorly to larger instances (Kwon et al., 2020; Luo et al., 2023): typically, the policy is trained with REINFORCE (Williams, 1992), where gradients are accumulated over full trajectories. This results in high memory requirements; hence, the autoregressive decoder of the underlying architecture is usually lightweight. Luo et al. (2023) and Drakulic et al. (2023) recently attributed the poor generalization ability to exactly the lightweight decoder of the used architectures (Drakulic et al., 2023; Luo et al., 2023). They show that increasing the size of the decoder structure and moving towards a decoder-only architecture addresses this issue and leads to superior generalization performance. However, the network's size makes it impractical to train the policy with the commonly used RL methods.

To be able to train these larger architectures without expert data, we propose in this paper a simple, problem-independent training scheme on the intersection of RL and SL: We decode a set of solutions for randomly generated problem instances in each epoch using the current model. The solution with the best objective function evaluation for each instance is treated as a *pseudo-expert* solution (corresponding to a *sequence* of incremental decisions). Like decoder-only models in natural language processing (NLP), we train the policy on these pseudo-expert solutions in a next-token prediction manner. By repeating this process, we create a 'self-improving' loop. While self-improving models by training them on their own output have been proposed in the context of NLP (Huang et al., 2023), our training strategy constitutes a novel proposition for NCO.

The effectiveness of this strategy relies on the quality of the decoded solutions and the efficiency of the underlying sampling process. In particular, we want to sample as few sequences as possible to get better and better solutions in each epoch. While repeated sampling from the conditional distribution given by the model at each sequential step works in principle, it can result in a set of sequences that lacks diversity, contains numerous duplicates, and requires a large number of samples to find a solution that enables the network to improve. In this paper, we propose a sampling mechanism based on batch-wise Stochastic Beam Search (SBS) (Kool et al., 2019c) as introduced by (Shi et al., 2020). The mechanism draws samples without replacement in multiple rounds while maintaining a search tree. We suggest taking advantage of the batch- and round-wise mechanism to enhance its effectiveness. Given the sequences sampled in a single round, we estimate the expected value of the objective function and update the trie with the advantage of the sampled sequences, a strategy derived from a provable policy improvement operation. Furthermore, to balance the explore-exploit tradeoff, we couple this update strategy with Top-$p$ (nucleus) sampling (Holtzman et al., 2020). After each round, we gradually increase $p \leq 1$ to allow more unreliable sequences. Our method applies to any constructive neural CO problem, unlike sampling strategies that leverage problem specifics to diversify and improve the sampled solutions (Kwon et al., 2020).

On the Traveling Salesman Problem (TSP) and the Capacitated Vehicle Routing Problem (CVRP), we demonstrate that training state-of-the-art architectures with our introduced method Gumbeldore (GD)[1] can

---

[1]In homage to *Stochastic Beams and where to find them* (Kool et al., 2019c).

produce policies of comparable strength to those trained with expert trajectories from solvers. In addition, we train a transformer-based architecture for the classical Job Shop Scheduling Problem (JSSP) using our method. Our results outperform the current state of the art by a wide margin.

Our contributions are summarized as follows:

(i) We propose a novel 'self-improvement' training strategy for NCO on the intersection of RL and SL: In each epoch, we decode a set of solutions from the current policy and treat the best sequences found as pseudo-expert trajectories on which we train the policy in a next-token prediction manner.

(ii) For sequence decoding, we propose a novel method based on drawing the sequences in multiple rounds using batch-wise SBS. We develop a technique to make the underlying search tree more informed in each round by updating the sequence probabilities according to their advantage. Informally, we increase (decrease) the probability of better (worse) trajectories than expected. This update adds next to no computational overhead to the sampling method, and we derive this technique from a provable policy improvement.

(iii) We show on the TSP and CVRP that our self-improvement method achieves comparable results to direct training on (near) optimal expert trajectories.

(iv) We present a novel transformer-based architecture for the JSSP that outperforms the current state of the art when trained with our method.

Our code for the experiments, data, and trained network weights are available at https://github.com/grimmlab/gumbeldore.

## 2 Related Work

**Learning constructive heuristics**   Initiated by Vinyals et al. (2015) and Bello et al. (2016), the constructive approach in neural CO uses a neural network (policy) to incrementally build solutions by selecting one element at a time. The most prominent way to train the policy network without relying on labeled data has become policy gradient methods in RL, notably using REINFORCE (Williams, 1992) with self-critical training (Rennie et al., 2017), where the result of a greedy rollout is used as a baseline for the gradient (Kool et al., 2019b). A large body of research focuses on keeping the network architecture fixed and instead improving the self-critical baseline and gradient estimation by *diversifying* the sampled solutions (Kool et al., 2019a; Kwon et al., 2020; Kool et al., 2020; Kim et al., 2021; 2022), often by exploiting problem-specific symmetries. For example, POMO (Kwon et al., 2020), one of the strongest constructive approaches, achieves diversity in routing problems by rolling out the model from all possible starting nodes of an instance. For the policy network, the Transformer (Vaswani et al., 2017) has become the standard choice for many CO problems (Deudon et al., 2018; Kool et al., 2019b; Kwon et al., 2020; Zhao et al., 2022) and is also used extensively in this work. However, current RL-based methods struggle to generalize to larger instances (Joshi et al., 2020), possibly due to their prevalent encoder-decoder structure, where a heavy encoder and light decoder facilitate training with policy gradient methods but limit scalability. On the other hand, recent work suggests that a lighter encoder with a heavier decoder can significantly improve generalization results (even with greedy inference only) (Drakulic et al., 2023; Luo et al., 2023), albeit at the cost of increased (if not prohibitive) computational complexity for training with RL.

**Self-improvement in neural CO**   The cross-entropy method (De Boer et al., 2005; Rubinstein, 1999) is a derivative-free optimization technique for a single problem instance that can be summarized as generating a set of solution candidates, evaluating them according to some objective function, and selecting the best-performing candidates to guide the generation of new solutions. While this principle of pushing the policy toward better performing sampled solutions lies at the heart of the self-critical policy gradient methods, there is little work on directly cloning the behavior of trajectories (from multiple instances) sampled from the current model in neural CO (Bogyrbayeva et al., 2022; Bengio et al., 2021). In fact, independent of our work, Corsini et al. (2024) make the same observation. They present a 'self-labeling' strategy that samples multiple solutions from the current policy and uses the best one as a pseudo-label for supervised training. However, they focus only on the JSSP and use vanilla sampling with replacement, in contrast to our main contribution, which updates the policy during the sampling without replacement process. We refer to Appendix E for a deeper

comparison. Furthermore, Luo et al. (2023) show a hybrid problem-specific way to learn without any labeled data by pre-training a randomly initialized model with RL on small instances, followed by reconstructing partial solutions to refine an unlabeled training dataset. We show that we can outperform this strategy without relying on RL.

**Improving at inference time**   Closely related to our proposed sampling methods are approaches that use inference-time search methods (other than greedy and pure beam search) to improve the generated solutions. While the aforementioned problem-specific diversification of POMO falls into this category, Choo et al. (2022) propose a beam search guided by greedy rollouts. Active Search (Bello et al., 2016) and EAS (Hottung et al., 2022) update (a subset of) the policy network parameters for individual instances using gradient descent. The tabular version of EAS is reminiscent of our approach by forcing sampled solutions to be close to the best solution found so far. MDAM (Xin et al., 2021) and Poppy (Grinsztajn et al., 2024) maintain a population of policies, while COMPASS (Chalumeau et al., 2024) learns a distribution of policies that is searched at test time. However, these search methods are designed only for test time to exploit a pre-trained model or require a significant computational budget on the order of minutes per instance. This makes them impractical in our self-improving setting, where we require trajectories that may live for only one epoch.

## 3   Learning Algorithm

In the following section, we formally set up the problem and present our proposed overarching self-improvement learning method.

### 3.1   Problem Setup

This paper considers a CO problem with $T \in \mathbb{N}$ decision variables of finite domain. Let $\mathcal{X}$ be some distribution over the problem *instances*. For each instance $x \sim \mathcal{X}$, we assume that we can construct a feasible solution sequentially by assigning a value $a_1$ to the first variable, then $a_2$ to the second, and so on, until we arrive at a complete trajectory $\boldsymbol{a}_{1:T} := (a_1, \ldots, a_T) \in \mathcal{S}_x$, where $\mathcal{S}_x$ is the set of all feasible solutions. Given an objective function $f_x \colon \mathcal{S}_x \to \mathbb{R}$ that maps a feasible solution to a real scalar, the goal is to find $\arg\max_{\boldsymbol{a}_{1:T} \in \mathcal{S}_x} f_x(\boldsymbol{a}_{1:T})$.

The corresponding neural sequence problem is to find a policy $\pi_\theta$ parameterized by $\theta$ and factorized in conditional distributions (i.e., the probability of a decision variable given the previous sequence), that maximizes the expectation $\mathbb{E}_{x \sim \mathcal{X}} \mathbb{E}_{\boldsymbol{a}_{1:T} \sim \pi_\theta} [f_x(\boldsymbol{a}_{1:T})]$, where we have the total probability $\pi_\theta(\boldsymbol{a}_{1:T}) = \prod_{t=1}^{T} \pi_\theta(a_t | \boldsymbol{a}_{1:t-1})$.

Here, for consistency of notation, we write $\boldsymbol{a}_{1:0}$ for an empty trajectory. We assume that we can sample from $\mathcal{X}$, and usually, $\mathcal{X}$ is chosen as a uniform distribution (e.g., for TSP, uniform distribution of points over the unit square from which we can sample random nodes). The policy $\pi_\theta$ is a *sequence model* that defines valid probability distributions over partial and complete sequences. In the following, we omit the parameter $\theta$ in the subscript and also refer to the decision variables as *tokens* by the terminology of sequence models. In particular, we use the terms 'sequence', 'trajectory' and 'solution' interchangeably.

### 3.2   Self-Improvement

We describe our proposed simple self-improving training scheme in Algorithm 1. We generate a set of random problem instances in each epoch and use the current best policy network to sample multiple trajectories for each instance. We add each instance and the corresponding best trajectory to the training dataset, from which the network is trained by imitation of the trajectories. In particular, we train the policy to predict the next token from a partial solution using a cross-entropy loss. At the end of the epoch, if the policy performs better when rolled out greedily than the currently best, we empty the dataset again (as we expect the sampled solutions to improve). Otherwise, the dataset will expand in the next epoch, providing the model with more training data.

The efficiency of the training is determined by the sampling method in line 7, highlighted in bold. The goal is to obtain a good solution with few samples $m$, balancing exploration and exploitation. Naively sampling sequences with replacement (i.e., Monte Carlo i.i.d. sampling, choosing the next token $a_t$ with probability

---

**Algorithm 1:** Self-improvement training for neural CO

---

**Input:** $\mathcal{X}$: distribution over problem instances; $f_*$: objective function
**Input:** $n$: number of instances to sample in each epoch
**Input:** $m$: number of sequences to sample for each instance
**Input:** VALIDATION $\sim \mathcal{X}$: validation dataset

**1** Randomly initialize policy $\pi_\theta$
**2** $\pi_{\text{best}} \leftarrow \pi_\theta$                                                                  ▷ current best policy
**3** DATASET $\leftarrow \emptyset$
**4** **foreach** epoch **do**
**5**     Sample set of $n$ problem instances INSTANCES $\sim \mathcal{X}$
**6**     **foreach** $x \in$ INSTANCES **do**
**7**         **Sample set of $m$ feasible solutions** $A := \{\boldsymbol{a}_{1:T}^{(1)}, \dots, \boldsymbol{a}_{1:T}^{(m)}\} \sim \pi_{\text{best}}$
**8**         DATASET $\leftarrow$ DATASET $\cup \left\{ \left(x, \arg\max_{\boldsymbol{a}_{1:T} \in A} f_x(\boldsymbol{a}_{1:T})\right)\right\}$          ▷ add best solution
**9**     **foreach** batch **do**
**10**         Sample $B$ instances and partial solutions $\boldsymbol{a}_{1:d_j}^{(j)} = (a_1^{(j)}, \dots, a_{d_j}^{(j)})$ with $d_j < T$ from DATASET for
             $j \in \{1, \dots, B\}$
**11**         Minimize batch-wise loss $\mathcal{L}_\theta = -\frac{1}{B}\sum_{j=1}^{B} \log \pi_\theta\left(a_{d_j+1}^{(j)} | \boldsymbol{a}_{1:d_j}^{(j)}\right)$          ▷ next-token prediction
**12**     **if** greedy performance of $\pi_\theta$ on VALIDATION better than $\pi_{\text{best}}$ **then**
**13**         $\pi_{\text{best}} \leftarrow \pi_\theta$                                                         ▷ update best policy
**14**         DATASET $\leftarrow \emptyset$                                                                  ▷ reset dataset

---

$\pi(a_t|\boldsymbol{a}_{1:t-1}))$ can lead to a homogeneous (often differing in a single token only (Li et al., 2016; Vijayakumar et al., 2018)) solution set with many duplicates. For example, Shi et al. (2020) show on a pre-trained TSP attention model (Kool et al., 2019b) that sampling 1280 sequences with replacement leads to $\sim$88% duplicate sequences for instances with 50 nodes, and still $\sim$17% duplicate sequences for instances with 100 nodes.

## 4    Sample Without Replacement, but Improvement

In this section, we describe our main contribution, a way to sample sequences without replacement in multiple rounds as described in Shi et al. (2020), coupled with a policy update after each round. We begin with a recall of Stochastic Beam Search (Kool et al., 2019c) using the Gumbel Top-k trick (Gumbel, 1954; Yellott Jr., 1977; Vieira, 2014), followed by a description of how to split the process into multiple rounds using an augmented trie, as described by Shi et al. (2020). We then describe how to combine the trie with a policy update. In the following, we use the term "beam search" to refer to the variant of beam search common in NLP, where nodes in the trie correspond to partial sequences, and a beam search of width $k$ keeps the top $k$ most probable sequences at each step, ranked by the total log-probability $\log \pi(\boldsymbol{a}_{1:t}) = \sum_{t'=1}^{t} \log \pi(a_{t'}|\boldsymbol{a}_{1:t'-1})$.

### 4.1    Preliminaries

**Stochastic Beam Search**    SBS is a modification of beam search that uses the Gumbel Top-k trick to sample sequences without replacement by using Gumbel *perturbed* log-probabilities. SBS starts at the root node (i.e., an empty sequence) $\boldsymbol{a}_{1:0}$ and sets the root perturbed log-probability $G_{\boldsymbol{a}_{1:0}} := 0$. Now, at any step during the beam search, let $\boldsymbol{a}_{1:t}$ be a partial sequence with score $G_{\boldsymbol{a}_{1:t}}$. Then, for a direct child $\boldsymbol{a}_{1:t+1} \in \text{Children}(\boldsymbol{a}_{1:t})$, we sample the perturbed log-probability $G_{\boldsymbol{a}_{1:t+1}} \sim \text{Gumbel}(\log \pi(\boldsymbol{a}_{1:t+1}))$ from a Gumbel distribution with location $\log \pi(\boldsymbol{a}_{1:t+1})$ *under the condition* that $\max_{\boldsymbol{a}_{1:t+1} \in \text{Children}(\boldsymbol{a}_{1:t})} G_{\boldsymbol{a}_{1:t+1}} = G_{\boldsymbol{a}_{1:t}}$, and use the $G_{\boldsymbol{a}_{1:t+1}}$'s as the scores for the beam search. By requiring that the maximum of the perturbed log-probabilities of sibling nodes must be equal to their parent's, sampled Gumbel noise is consistently propagated down the subtree. Kool et al. show that this modified beam search is equivalent to sampling $k$ sequences without replacement from the sequence model $\pi$, yielding diverse but high-quality sequences. Since SBS performs the expansion

of the beam as in regular deterministic beam search, it can be parallelized in the same way. We provide a detailed summary of the underlying theory in Appendix A.

**Incremental Sampling Without Replacement** Shi et al. (2020) present an *incremental* view on sampling sequences without replacement that can draw one sequence after another. They suggest maintaining an augmented trie structure that grows with the sampling process, where nodes correspond to partial sequences as in beam search. Starting from the root node (representing an empty sequence), a single leaf node $\boldsymbol{a}_{1:T}$ (representing a complete sequence) is sampled from the sequence model $\pi$. Then the total probability of $\pi(\boldsymbol{a}_{1:T})$ is recursively subtracted from the probability of the leaf and its ancestors (which correspond to all partial sequences $\boldsymbol{a}_{1:0}, \boldsymbol{a}_{1:1}, \ldots, \boldsymbol{a}_{1:T}$). After renormalization of the probabilities, the trie represents a sequence model from which $\boldsymbol{a}_{1:T}$ has been removed, and re-sampling from this updated trie is equivalent to sampling from the model conditioned on the fact that $\boldsymbol{a}_{1:T}$ cannot be selected (see Figure 1).

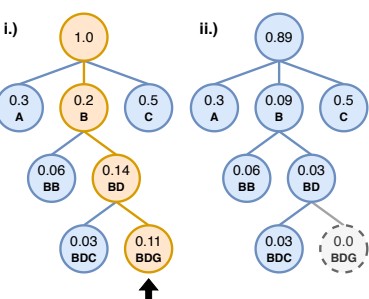

The incremental nature of the process has the advantage that we can sample continuously without replacement, stopping only when some condition is met. To further exploit the parallelism advantages of SBS, Shi et al. (2020) suggest combining SBS with their method: Instead of drawing just one sequence, we use the same trie for SBS and draw a batch of $k$ samples in parallel. Then, the recursive probability update is done for all $k$ sequences. This allows us to sample batches in *rounds*, e.g., with a beam width of $k$ and $n$ rounds, we sample $k \cdot n$ sequences without replacement. Note that although we must maintain a trie structure, each round of SBS maintains the constant memory requirements of a beam search of width $k$.

Figure 1: Example of incremental sampling using a trie. Nodes represent partial sequences (here, letters) and contain their total probabilities. i.) The sequence 'BDG' is sampled from the model, and we subtract its total probability from its ancestors. ii.) Updated trie after erasing 'BDG'.

## 4.2 Sampling with Gumbeldore

In the following, let $x \sim \mathcal{X}$ be a problem instance, and let $k$ be the beam width for SBS and $n$ the number of rounds. For sampling from a policy $\pi$ in Algorithm 1 above, we use the batch-wise sampling in $n$ rounds to draw $k \cdot n$ sequences without replacement. Suppose that in the first round we sample complete trajectories $\boldsymbol{a}_{1:T}^{(1)}, \ldots, \boldsymbol{a}_{1:T}^{(k)}$ with SBS. The question arises since we already have $k$ complete trajectories, and thus their evaluations $f_x(\boldsymbol{a}_{1:T}^{(i)}) \in \mathbb{R}$, can we get information about their quality and use it in the next round of SBS? Intuitively, for the next round of sampling, we would like to move in the trie from $\pi$ to a policy $\pi'$ that puts more emphasis on tokens that have led to 'good' solutions and less emphasis on tokens that have led to 'not-so-good' solutions.

### 4.2.1 Trie Update

**Theoretical Policy Improvement** To theoretically motivate our approach, consider *any* complete trajectory $\boldsymbol{a}_{1:T} = (a_1, \ldots, a_T)$ and a policy $\pi$. We can move to an improved policy $\pi'$ by shifting the unnormalized log-probability ('logit') of the intermediate tokens of $\boldsymbol{a}_{1:T}$ by their individual *advantage*. That is, for $i \in \{1, \ldots, T\}$, we set

$$\text{logit } \pi'(a_i | \boldsymbol{a}_{1:i-1}) := \log \pi(a_i | \boldsymbol{a}_{1:i-1}) +$$

$$\sigma \cdot \left( \underbrace{\mathbb{E}_{\boldsymbol{a}'_{1:T} \sim \pi(\cdot | \boldsymbol{a}_{1:i})} [f_x(\boldsymbol{a}'_{1:T})] - \mathbb{E}_{\boldsymbol{a}'_{1:T} \sim \pi(\cdot | \boldsymbol{a}_{1:i-1})} [f_x(\boldsymbol{a}'_{1:T})]}_{\text{advantage of token } a_i} \right). \tag{1}$$

Here, $\sigma > 0$ is a predefined size of the update step, and $\boldsymbol{a}'_{1:T} \sim \pi(\cdot | \boldsymbol{a}_{1:i})$ denotes a complete sequence $\boldsymbol{a}'_{1:T} = (a'_1, \ldots, a'_T)$ drawn from $\pi$ where $a'_j = a_j$ for all $j \in \{1, \ldots, i\}$. Informally, (1) increases the probability of tokens that have a better expected outcome than their parent and decreases it otherwise. Formally, (1) yields a policy improvement, i.e., we get

$$\mathbb{E}_{\boldsymbol{a}_{1:T} \sim \pi'} [f_x(\boldsymbol{a}_{1:T})] \geq \mathbb{E}_{\boldsymbol{a}_{1:T} \sim \pi} [f_x(\boldsymbol{a}_{1:T})], \tag{2}$$

and we provide a proof of this in Appendix B.

**Practical policy update**   Let $\boldsymbol{a}^{(1)}, \ldots, \boldsymbol{a}^{(k)}$ be $k$ complete trajectories sampled with SBS, which we assume to be ordered by their perturbed log-probabilities $G_{\boldsymbol{a}^{(1)}} \geq \cdots \geq G_{\boldsymbol{a}^{(k)}}$. We have omitted $1:T$ in the subscript for ease of notation. Motivated by its provable improvement, we would like to apply (1) to all $\boldsymbol{a}^{(i)}$, but the update relies on our ability to estimate the corresponding expectations. Kool et al. (2019c) present an unbiased estimator of the objective function expectation of the sequence model:

$$\mathbb{E}_{\boldsymbol{a}_{1:T} \sim \pi}\left[f_x(\boldsymbol{a}_{1:T})\right] \approx \sum_{i=1}^{k-1} w_{\boldsymbol{a}^{(k)}}(\boldsymbol{a}^{(i)}) f_x(\boldsymbol{a}^{(i)}), \text{ where } w_{\boldsymbol{a}^{(k)}}(\boldsymbol{a}^{(i)}) = \frac{\pi(\boldsymbol{a}^{(i)})}{q_{\boldsymbol{a}^{(k)}}(\boldsymbol{a}^{(i)})}, \tag{3}$$

and $q_{\boldsymbol{a}^{(k)}}(\boldsymbol{a}^{(i)}) = P(G_{\boldsymbol{a}^{(i)}} > G_{\boldsymbol{a}^{(k)}})$ is the probability that the $i$-th perturbed log-probability is greater than the $k$-th largest. In practice, the normalized variant (which is biased but consistent)

$$\mu(\boldsymbol{a}^{(1)}, \ldots, \boldsymbol{a}^{(k)}) := \frac{\sum_{i=1}^{k-1} w_{\boldsymbol{a}^{(k)}}(\boldsymbol{a}^{(i)}) f_x(\boldsymbol{a}^{(i)})}{\sum_{i=1}^{k-1} w_{\boldsymbol{a}^{(k)}}(\boldsymbol{a}^{(i)})} \tag{4}$$

is preferred to reduce variance. Note that (4) estimates the 'full' expectation from an empty sequence. However, since in SBS, the maximum of the perturbed log-probabilities of sibling nodes is conditioned on their parent, we can also estimate the expectation starting from other partial sequences: For any sequence $\boldsymbol{b}_{1:j}$ of length $j$, let $S(\boldsymbol{b}_{1:j}) := \{\boldsymbol{a}^{(m_1)}, \ldots, \boldsymbol{a}^{(m_l)}\} \subseteq \{\boldsymbol{a}^{(1)}, \ldots, \boldsymbol{a}^{(k)}\}$ (again ordered by their perturbed log-probabilities) be the subset of all sampled sequences that share the subsequence $\boldsymbol{b}_{1:j}$. I.e., we have $\boldsymbol{b}_{1:j} = \boldsymbol{a}_{1:j}^{(m_1)} = \cdots = \boldsymbol{a}_{1:j}^{(m_l)}$. Then,

$$\mathbb{E}_{\boldsymbol{a}_{1:T} \sim \pi(\cdot|\boldsymbol{b}_{1:j})}\left[f_x(\boldsymbol{a}_{1:T})\right] \approx \mu(\boldsymbol{a}^{(m_1)}, \ldots, \boldsymbol{a}^{(m_l)}) \tag{5}$$

estimates the expectation of the objective function given $\boldsymbol{b}_{1:j}$. While we can use this approach to apply (1) to the logits of any $\boldsymbol{a}^{(i)}$, it has a major drawback: Depending on the length $T$ of the problem, the beam width $k$, and the confidence of $\pi$, in practice the sampled sequences will rarely share a long subsequence, which makes it hard to properly estimate the expectation of nodes deep in the trie.

So we take a practical approach and *only* use the 'global' advantage $f_x(\boldsymbol{a}^i) - \mu(\boldsymbol{a}^{(1)}, \ldots, \boldsymbol{a}^{(k)})$, which we propagate up the trie at the same time as we mark $\boldsymbol{a}^{(i)}$ as sampled. Overall, the trie update after each round of sampling can then be summarized as follows: For all $i \in \{1, \ldots, k\}$, we compute the global advantage $A_{\boldsymbol{a}^{(i)}} := f_x(\boldsymbol{a}^{(i)}) - \mu(\boldsymbol{a}^{(1)}, \ldots, \boldsymbol{a}^{(k)})$, and update the logit for each $j \in \{1, \ldots, T\}$ via

$$\text{logit } \pi'(a_j^{(i)}|\boldsymbol{a}_{1:j-1}^{(i)}) := \log\left(\pi(a_j^{(i)}|\boldsymbol{a}_{1:j-1}^{(i)}) - \sum_{\boldsymbol{b} \in S(\boldsymbol{a}_{1:j}^{(i)})} \pi(\boldsymbol{b})\right) + \sigma \sum_{\boldsymbol{b} \in S(\boldsymbol{a}_{1:j}^{(i)})} A_{\boldsymbol{b}}. \tag{6}$$

We illustrate an example of the trie update after each round in Figure 2. We note that as all expectations are obtained via (4), the update (6) adds no significant computational overhead to the round-based SBS. When samling another $k$ sequences in the next round, we repeat the process. In Appendix C.1, we experimentally compare the theoretical update (1) with the practical update (6), and see that while both yield strong results, (6) is almost always more favorable in practice.

**Choice of step size** $\sigma$   The update (6) scales the probabilities by $\exp(\sigma \sum A_{\boldsymbol{b}})$, so the role of the hyperparameter $\sigma$ is mainly to dampen the advantage. Since the magnitude of $\sum A_{\boldsymbol{b}}$ usually also changes as the policy $\pi$ evolves, the choice of $\sigma$ is highly problem-dependent. In particular, one could think of, e.g., transforming the $A_{\boldsymbol{b}}$ by min-max normalization or changing $\sigma$ during training to optimize the behavior of the update. However, our experiments use the pure advantages and a constant, small problem-dependent $\sigma$ throughout the training process (see Section 5.3).

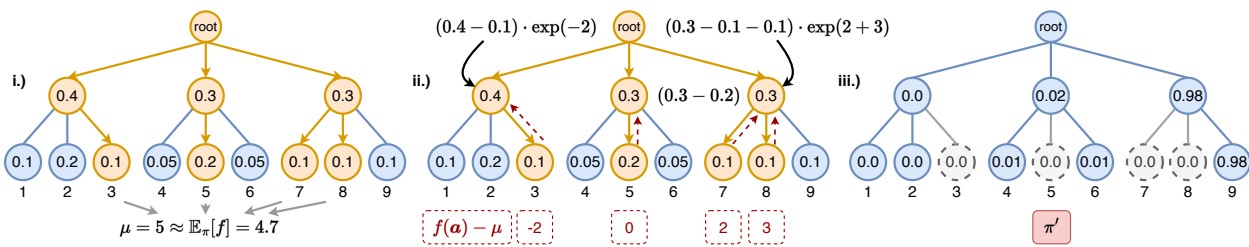

Figure 2: Example of update (6) with beam width $k = 3$ and $\sigma = 1$. Nodes represent partial sequences and their total probabilities; the numbers $\{1, \ldots, 9\}$ below the leaf nodes are their corresponding objective function evaluation. i.) We sample $k$ leaf nodes with SBS and obtain the objective function estimate $\mu$ with (4). ii.) We mark each leaf as sampled by removing its total probability from its ancestors. At the same time (equivalent to adding to the log-probability), we multiply the ancestors' reduced probability by the exponential of the leaf's advantage. iii.) Trie after normalization, corresponding to $\pi'$.

---

**Algorithm 2:** Sampling with Gumbeldore (GD)

**Input:** $x \sim \mathcal{X}$: problem instance; $f = f_x$: objective function; $\pi$: policy
**Input:** $k$: beam width for SBS; $n$: number of rounds
**Input:** $\sigma$: policy update step size; $p_{\min}$: minimum nucleus size
**Output:** Sampled sequence with highest $f$

1   SAMPLED $\leftarrow \emptyset$
2   **for** $i = 1, \ldots, n$ **do**
3     $p \leftarrow (1 - \frac{i-1}{n-1}) \cdot p_{\min} + \frac{i-1}{n-1}$
4     $\boldsymbol{a}_{1:T}^{(1)}, \ldots, \boldsymbol{a}_{1:T}^{(k)} \leftarrow$ SBS with $\pi$, beam width $k$ and nucleus $p$
5     SAMPLED $\leftarrow$ SAMPLED $\cup \{\boldsymbol{a}_{1:T}^{(1)}, \ldots, \boldsymbol{a}_{1:T}^{(k)}\}$
6     Obtain $\mathbb{E}_\pi[f]$ estimate $\mu \leftarrow \mu(\boldsymbol{a}^{(1)}, \ldots, \boldsymbol{a}^{(k)})$ via (4)
7     $\pi' \leftarrow \pi$
8     **for** $j = 1, \ldots, k$ **do**
9       **for** $m = 1, \ldots, T$ **do**
10        $S \leftarrow S(\boldsymbol{a}_{1:m}^{(j)}) \subseteq \{\boldsymbol{a}_{1:T}^{(1)}, \ldots, \boldsymbol{a}_{1:T}^{(k)}\}$         ▷ `sequences with same first` $m$ `tokens`
11        logit $\pi'(a_m^{(j)}|\boldsymbol{a}_{1:m-1}^{(j)}) \leftarrow \log\left(\pi(a_m^{(j)}|\boldsymbol{a}_{1:m-1}^{(j)}) - \sum_{\boldsymbol{b} \in S} \pi(\boldsymbol{b})\right) + \sigma \sum_{\boldsymbol{b} \in S}(f(\boldsymbol{b}) - \mu)$

12   Return $\arg\max_{\boldsymbol{a} \in \text{SAMPLED}} f(\boldsymbol{a})$

---

### 4.2.2   Growing Nucleus

To further exploit the round-based sampling strategy, we truncate the conditional distributions $\pi(\cdot|\boldsymbol{a}_{1:t})$ using Top-$p$ (nucleus) sampling (Holtzman et al., 2020) at each beam expansion in SBS with *varying p*. If $n > 1$ is the number of rounds, then we set $p$ for the $i$-th round with $1 \leq i \leq n$ to $p = \left(1 - \dfrac{i-1}{n-1}\right) \cdot p_{\min} + \dfrac{i-1}{n-1}$. We start at a predefined $0 < p_{\min} < 1$ and linearly increase $p$ to 1. The idea is that truncating the "unreliable tail" of $\pi(\cdot|\boldsymbol{a}_{1:t})$ with $p < 1$ may be desirable in later stages of training to further exploit the model. However, if the model is unreliable, especially in the early stages, we want to consider the full distribution to encourage exploration. By gradually increasing $p$, we accommodate both cases, while the trie policy update helps keep good choices in the nucleus and pushes bad ones out. In practice, we start with $p_{\min} = 1$ and set $p_{\min} = 0.95$ after several epochs. When sampling at inference time, an even smaller $p_{\min}$ can improve the results (cf. Appendix C.3). During training, however, we found that using $p_{\min} < 0.9$ leads to the model converging prematurely as it only learns to amplify its decisions.

We summarize the complete sampling strategy in Algorithm 2.

# 5 Experimental Evaluation

We are pursuing two experimental goals: First, we want to evaluate the effectiveness of our method within the entire training cycle of the self-improvement strategy (see Algorithm 1), and we test it on three CO problems with different underlying network architectures: the two-dimensional Euclidean TSP, the CVRP, and the standard JSSP. Second, as the training relies on the effectiveness of the solution sampling, we want to assess the solution quality of our sampling strategy (especially with a low number of samples). We take various network checkpoints of all three problems and compare different sampling techniques with our proposed method.

## 5.1 Routing Problems

We consider two prevalent routing problems, the two-dimensional Euclidean TSP and CVRP, where the constructive approach builds a tour by picking one node after another. For both problems, we reimplement the transformer-based architecture from the recent Bisimulation Quotienting (BQ) method (Drakulic et al., 2023). This model has nine layers that process the remaining nodes at each step. In the original work, due to its size, BQ is trained on expert trajectories from solvers with $N = 100$ nodes. The authors report excellent results on the training distribution and superior generalization on larger graphs with $N \in \{200, 500, 1000\}$ nodes. We train the network on instances with 100 nodes in the same supervised manner. In addition, we train it with two variants of our proposed self-improvement method, where we sample trajectories without replacement using **(a)** simple round-wise SBS (SI WOR), and **(b)** the proposed Gumbeldore method (SI GD). The setup is briefly outlined below. For implementation details and hyperparameters, please refer to Appendix D.2 and D.3. Furthermore, the 'Light Encoder Heavy Decoder' (LEHD) by (Luo et al., 2023) follows a similar paradigm to BQ. Apart from supervised learning, they show a problem-specific way to learn on the TSP without any labeled data. We discuss this method in Appendix C.2 and demonstrate that SI GD outperforms it.

**Datasets and supervised training** Training (for supervised learning), validation, and test data are generated in the standard way of previous work (Kool et al., 2019b) by uniformly sampling $N$ points from the two-dimensional unit square. The training dataset consists of one million randomly generated instances with $N = 100$ nodes for TSP. The validation and test sets (also $N = 100$) include 10,000 instances each. The test set is the same widely used dataset generated by Kool et al. (2019b). For $N \in \{200, 500, 1000\}$, we use a test set of 128 instances identical to Drakulic et al. (2023). For supervised training and to compute optimality gaps, we obtain optimal solutions for the generated instances with the Concorde solver (Applegate et al., 2006). For the CVRP, the datasets have the same number of instances, and we use the same sets as in Luo et al. (2023), where solutions come from HGS (Vidal, 2022). The vehicle capacities for 100, 200, 500, and 1000 nodes are 50, 80, 100, and 250, respectively. The supervised model is trained by imitation of the expert solutions using a cross-entropy loss. We sample 1000 batches of 1024 sub-paths in each epoch, following Drakulic et al. (2023). The model is trained until we have yet to observe any improvement on the validation set for 100 epochs (in total ∼4.5k epochs for TSP and ∼1.3k for CVRP). We note that we can reproduce (up to fluctuations) the results from the original work.

**Self-improvement training** For both SI WOR and SI GD, we generate 1,000 random instances in each epoch. Per instance, we sample 128 solutions in $n = 4$ rounds with a beam width of $k = 32$. For SI GD, we set the constant $\sigma$ to scale the advantages to $\sigma = 0.3$ for the TSP and to $\sigma = 3$ for the CVRP. We refer to Appendix D.1 for how we obtained these values. For SI GD and both routing problems, we start with $p_{\min} = 1$ (no nucleus sampling) and set $p_{\min} = 0.95$ after 500 epochs. Training on the generated data is performed in the same supervised way (in total ∼2.2k epochs for TSP and ∼2.7k for CVRP). Interestingly, SI GD takes fewer epochs on the TSP and more on the CVRP than SL, confirming that it is much harder to learn heuristics from scratch for the CVRP. Our method cannot access optimality information, so we compare model checkpoints by their average tour length on the validation set.

**Inference and baselines** All evaluations are performed with the model trained on $N = 100$. We are mainly interested in how the self-improving model performs compared to its supervised counterpart (BQ SL)

Table 1: Results for the routing problems, with emphasis on the BQ architecture. 'Gap' is the optimality gap w.r.t Concorde (Applegate et al., 2006) for the TSP and HGS (Vidal, 2022) for the CVRP. 'Time' is the time needed to solve all instances. Models are trained on instances with $N = 100$ nodes and used to evaluate generalization to $N \in \{200, 500, 1000\}$ nodes. 'beam' refers to beam search of given width.

| Method | Test (10k inst.) | | Generalization (128 instances) | | | | | |
| --- | --- | --- | --- | --- | --- | --- | --- | --- |
| | TSP $N = 100$ | | TSP $N = 200$ | | TSP $N = 500$ | | TSP $N = 1000$ | |
| | Gap | Time | Gap | Time | Gap | Time | Gap | Time |
| OR-Tools | 2.90% | 1h | 4.52% | 6m | 4.89% | 30m | 5.02% | 3h |
| AM, beam 1024 | 2.49% | 5m | 6.18% | 15s | 17.98% | 2m | 29.75% | 7m |
| MDAM, beam 50 | 0.40% | 20m | 2.04% | 3m | 9.88% | 12m | 19.96% | 1h |
| POMO, augx8 | 0.14% | 15s | 1.57% | 2s | 20.18% | 16s | 40.60% | 3m |
| SGBS | 0.06% | 8m | 0.67% | 30s | 11.42% | 10m | 25.25% | 1.5h |
| BQ SL, greedy | 0.40% | 30s | 0.60% | 3s | 0.98% | 16s | 1.72% | 32s |
| BQ SL, beam 16 | 0.02% | 8m | 0.09% | 30s | 0.43% | 4m | 0.91% | 10m |
| **BQ SI WOR (ours)**, greedy | 0.44% | 30s | 0.73% | 3s | 1.39% | 16s | 2.34% | 32s |
| **BQ SI WOR (ours)**, beam 16 | 0.03% | 8m | 0.11% | 30s | 0.53% | 4m | 1.16% | 10m |
| **BQ SI GD (ours)**, greedy | 0.41% | 30s | 0.64% | 3s | 1.12% | 16s | 2.11% | 32s |
| **BQ SI GD (ours)**, beam 16 | 0.02% | 8m | 0.10% | 30s | 0.46% | 4m | 1.01% | 10m |
| | CVRP $N = 100$ | | CVRP $N = 200$ | | CVRP $N = 500$ | | CVRP $N = 1000$ | |
| | Gap | Time | Gap | Time | Gap | Time | Gap | Time |
| OR-Tools | 6.19% | 2h | 6.89% | 1h | 9.11% | 2h | 11.66% | 3h |
| AM, beam 1024 | 4.20% | 10m | 8.18% | 24s | 18.01% | 3m | 87.56% | 12m |
| MDAM, beam 50 | 2.21% | 25m | 4.30% | 3m | 10.50% | 12m | 27.81% | 47m |
| POMO, augx8 | 0.69% | 25s | 4.87% | 3s | 19.90% | 24s | 128.89% | 4m |
| SGBS | 0.08% | 20m | 2.58% | 50s | 15.34% | 12m | 136.98% | 2h |
| BQ SL, greedy | 3.03% | 0s | 2.63% | 4s | 3.75% | 22s | 5.30% | 48s |
| BQ SL, beam 16 | 1.22% | 13m | 1.15% | 1m | 1.93% | 6m | 2.49% | 15m |
| **BQ SI WOR (ours)**, greedy | 3.93% | 50s | 4.11% | 4s | 4.87% | 22s | 8.47% | 48s |
| **BQ SI WOR (ours)**, beam 16 | 2.06% | 13m | 2.06% | 1m | 2.91% | 6m | 5.87% | 15m |
| **BQ SI GD (ours)**, greedy | 3.26% | 50s | 3.05% | 4s | 3.89% | 22s | 8.33% | 48s |
| **BQ SI GD (ours)**, beam 16 | 1.72% | 13m | 1.58% | 1m | 2.32% | 6m | 5.57% | 15m |

and thus focus on greedy results. We also report the results using a beam search of width 16 by Drakulic et al. (2023) for coherence. Furthermore, we list the results of **(a)** Google's OR-Tools (Perron & Furnon, 2022) as a non-learning local search algorithm; **(b)** the prominent Attention Model (AM) (Kool et al., 2019b) (with a beam search of width 1024), which was trained with self-critical REINFORCE and forms the basis of most constructive methods for routing; **(c)** its multi-decoder version MDAM (Xin et al., 2021) (with a beam search of width 50), which trains multiple diverse policies; **(d)** POMO (Kwon et al., 2020) with its best inference technique (corresponding to $8N$ diversified solutions per instance of size $N$). **(e)** SGBS (Choo et al., 2022), a pure inference technique based on beam search guided by greedy rollouts, backed by POMO with hyperparameters $(\beta, \gamma)$ which are set to $(10, 10)$ for the TSP and $(4, 4)$ for the CVRP. POMO is considered the current state-of-the-art method of learning constructive heuristics from scratch. POMO performs impressively on the training distribution but has difficulties generalizing to larger instances. As our architectural setup is identical to BQ, we refer to the original work by (Drakulic et al., 2023) for comparison with various other methods.

**Results** Table 1 summarizes the optimality gaps. Both SI WOR and SI GD show strong performance when compared to training on expert trajectories, with SI GD outperforming SI WOR. For the TSP, SI GD achieves results on par with its SL counterpart on the training distribution $N = 100$ and shows similarly strong generalization capabilities. For the CVRP, while the results are close to SL on $N = 100$, the generalization

difference becomes larger as $N$ grows, with, for example, a difference of $\sim 3\%$ on $N = 1000$. We note that generalizing to larger instances means not only larger $N$ but also unseen vehicle capacities. Drakulic et al. (2023) analyze that the final performance of the model strongly depends on how an expert solution sorts the subtours. In particular, the model favors learning from solutions where the vehicle starts with subtours with the smallest remaining vehicle capacity at the end of the subtour (usually 0). Since we cannot guarantee during learning in our SI procedure that the vehicle is nearly optimally utilized in the subtours, we assume that this makes it more difficult for the model to generalize to unknown capacities. Nevertheless, the SI GD and SI WOR generalization results for CVRP still show the same strong dynamics as those of BQ SL, trained without any labeled data. This makes our method a promising alternative to RL-based policy gradient methods, especially due to its simplicity.

## 5.2 Job-Shop Scheduling Problem

In a JSSP instance of size $J \times M$, we are given $J$ jobs consisting of $M$ operations. Each job's operations must be processed in order (called a precedence constraint), and each operation runs on exactly one of $M$ machines for a given processing time (i.e., there is a bijection between a job's set of operations and the set of machines). The goal is to find a schedule that processes all job operations without violating the job's precedence constraints and has a minimum makespan (i.e., the completion time of the last operation). In our required sequential language, we represent a feasible solution, i.e., a schedule that satisfies all precedence constraints, as a *sequence of jobs*, where each occurrence of a job means scheduling the next unscheduled operation of the job at the earliest possible time. In particular, the model predicts a probability distribution over all unfinished jobs at each step of constructing a solution.

**Model**  We propose a novel pure transformer-based architecture inspired by the BQ principle (Drakulic et al., 2023) to train a model with our SI GD method. We consider a JSSP instance as an unordered sequence of operations with positional information for operations belonging to the same job. We mask already scheduled operations at each construction step and process the sequence through a simple stack of *pairs* of attention layers. In each pair, we mask operations so that i.) in the first layer, only operations that belong to the *same job* can attend to each other, and ii.) in the second layer, only operations that need to run on the *same machine* can attend to each other. We describe the architecture in detail in Appendix D.4. To balance inference speed and model expressiveness, we settle on a latent dimension of 64, three pairs of attention layers with eight heads, and a feed-forward dimension of 256.

**Datasets**  All random instances are generated in the standard manner of Taillard (1993) with integer processing times in $[1, 99]$. Since we train our model only with our self-improvement method, we pre-generate a validation set of 100 instances of size $20 \times 20$. We test the model on the well-known Taillard benchmark dataset (Taillard, 1993), computing optimality gaps to the best-known upper bounds.[2]

**Self-improvement training**  We train the policy for 100 epochs with SI WOR and SI GD. In each epoch, we randomly choose a size from $\{15 \times 10, 15 \times 15, 15 \times 20\}$ and sample 512 instances of the chosen size. We generate 128 solutions for each instance in 4 rounds of beam width 32. For SI GD, we use an advantage step size of $\sigma = 0.05$ and set $p_{\min} = 0.95$ after 50 epochs. Similar to the routing problems, we sample 1,000 incomplete schedules (i.e., a partial sequence of jobs) of size 512 from the generated training data. We train the model to predict the next job to choose using a cross-entropy loss. Please see Appendix D.4 for more details.

**Inference and baselines**  We test performance of the model by unrolling the policy greedily. Since SI GD shows the strongest greedy results, we further evaluate the trained model i.) with a beam search of width 16, ii.) SBS of beam width 16 with a constant nucleus of $p = 0.8$ and iii.) sampling with GD to assess the efficiency of GD as an inference technique. We compare our approach with **(a)** L2D (Zhang et al., 2020), an RL single-agent method; **(b)** ScheduleNet (Park et al., 2022), an RL multi-agent method. Both methods represent a JSSP instance as a disjunctive graph and train Graph Neural Networks (GNNs) with (variants of) PPO (Schulman et al., 2017); **(c)** The 500- and 5,000-step variants of L2S (Zhang et al., 2024),

---

[2]Available at `http://optmizizer.com/TA.php` and `http://jobshop.jjvh.nl`

Table 2: JSSP results on the Taillard benchmark set with eight different problem sizes. Our models SI WOR and SI GD are trained on instances of random sizes $\in \{15 \times 10, 15 \times 15, 15 \times 20\}$, using the resulting model for all benchmark instances. The first group of results represents greedy inference, while the second group compares inference types with larger computation times. 'GD $32 \times 4$' refers to using GD sampling with four rounds of beam width 32, using $p_{\min} = 0.8$. 'SBS 16, $p = 0.8$' refers to sampling 16 sequences without replacement with a constant nucleus size of $p = 0.8$. For GD $32 \times 4$ and SBS 16, we report the mean across five repetitions. Best optimality gaps per group are indicated in bold.

| Method | 15 × 15 Gap | Time | 20 × 15 Gap | Time | 20 × 20 Gap | Time | 30 × 15 Gap | Time | 30 × 20 Gap | Time | 50 × 15 Gap | Time | 50 × 20 Gap | Time | 100 × 20 Gap | Time |
|---|---|---|---|---|---|---|---|---|---|---|---|---|---|---|---|---|
| CP-SAT | 0.1% | 1.25h | 0.2% | 8h | 0.7% | 10h | 2.1% | 10h | 2.8% | 10h | 0.0% | 24m | 2.8% | 10h | 3.9% | 10h |
| L2D, greedy | 26.0% | 0s | 30.0% | 0s | 31.6% | 1s | 33.0% | 1s | 33.6% | 2s | 22.4% | 2s | 26.5% | 4s | 13.6% | 25s |
| ScheduleNet, greedy | 15.3% | 3s | 19.4% | 6s | 17.2% | 11s | 19.1% | 15s | 23.7% | 25s | 13.9% | 50s | 13.5% | 1.6m | 6.7% | 7m |
| SPN, greedy | 13.8% | 0s | 15.0% | 0s | 15.2% | 0s | 17.1% | 0s | 18.5% | 1s | 10.1% | 1s | 11.6% | 1s | 5.9% | 2s |
| L2S, 500 steps | **9.3%** | 9s | 11.6% | 10s | 12.4% | 11s | 14.7% | 12s | 17.5% | 14s | 11.0% | 16s | 13.0% | 23s | 7.9% | 50s |
| SI WOR (ours), greedy | 12.1% | 1s | 10.9% | 1s | 11.5% | 1s | 10.7% | 1s | **13.6%** | 2s | 3.9% | 2s | 7.9% | 2s | 1.7% | 28s |
| SI GD (ours), greedy | 9.6% | 1s | **9.9%** | 1s | **11.1%** | 1s | **9.5%** | 1s | 13.8% | 2s | **2.7%** | 2s | **6.7%** | 3s | **1.7%** | 28s |
| L2S, 5000 steps | 6.2% | 1.5m | 8.3% | 1.7m | 9.0% | 1.9m | 9.0% | 2m | 12.6% | 2.4m | 4.6% | 2.8m | 6.5% | 3.8m | 3.0% | 8.4m |
| SI GD (ours), beam 16 | 10.1% | 2s | 9.8% | 2s | 10.4% | 4s | 8.5% | 5s | 12.3% | 10s | 2.6% | 18s | 7.7% | 40s | 1.3% | 1.5m |
| SI GD, SBS 16, $p = 0.8$ | 6.0% | 2s | 6.6% | 2s | 8.9% | 4s | 7.4% | 5s | 10.5% | 10s | 1.7% | 18s | 5.0% | 40s | 0.7% | 1.5m |
| SI GD, GD $32 \times 4$ | **5.0%** | 4s | **5.6%** | 10s | **7.0%** | 30s | **5.8%** | 36s | **9.2%** | 1.3m | **1.1%** | 2.4m | **4.3%** | 3.5m | **0.4%** | 9m |

a strong GNN-guided improvement method that transitions between complete solutions at each step; **(d)** the concurrent work SPN (Corsini et al., 2024), which trains a Graph Attention Network in a self-improving manner on sampled solutions (we refer to Appendix E for a methodological comparison); and **(d)** as a non-learning baseline, the highly efficient constraint programming solver CP-SAT in OR-Tools (Perron & Furnon, 2022), with a time limit of one hour per instance, as reported by Zhang et al. (2024). As noted in most previous work on neural CO, a fair comparison of inference times is challenging, as they depend on the implementation (especially using deep learning frameworks) by orders of magnitude. In particular, we report the time required to solve the instances of a given size (allowing parallelization) but compare the respective methods based on their inference type (e.g., treating greedy methods as equally computationally intensive).

**Results**  We collect the results in Table 2. In the greedy setting, both SI GD and SI WOR outperform all three constructive methods, L2D, ScheduleNet, and SPN, by a wide margin and obtain smaller gaps than L2S with 500 improvement steps in all but one case. The quadratic complexity of our architecture becomes noticeable for $100 \times 20$. SI GD significantly improves SI WOR. When allowing for longer inference times, we compare our strongest model SI GD only with the strongest method, L2S with 5,000 improvement steps. We note that our method does not always improve with beam search, with some results even worse than greedy. That sequences with high overall probability do not necessarily mean high quality is a ubiquitous effect in NLP (Holtzman et al., 2020), showing that the model is not as confident as for the routing problems. Thus, we use the model obtained with SI GD and *sample* with SBS and a constant nucleus of $p = 0.8$. This maintains short inference times but outperforms L2S across all problem sizes. We can further improve the results by sampling with GD in multiple rounds (with $p_{\min} = 0.8$) and using our policy update. For more results and comparisons when using GD at inference time, see Appendix C.3.

## 5.3 Sampling Performance

We analyze the performance of GD as a sampling technique, especially when sampling only a small number of sequences. For this, we take the model weights from two checkpoints during training: One late in training, when the model has almost converged, and an intermediate one, when the model shows a fair greedy performance but still has much room for improvement. Taking a checkpoint very early in training carries limited information, as the performance still depends strongly on weight initialization, and any sampling method with high exploration leads to substantial improvement. We then sample up to 640 sequences with an SBS beam width of 32 in up to $n = 20$ rounds (for sampling with replacement, we sample the equivalent number of $32 \cdot n$ sequences) and compare the optimality gap of the best sequences found. In Figure 3, we plot the results using our proposed method ('Gumbeldore') and for sampling with ('Sample WR') and without replacement ('Sample WOR'). To better understand the impact of the growing nucleus in Section 4.2.2, we

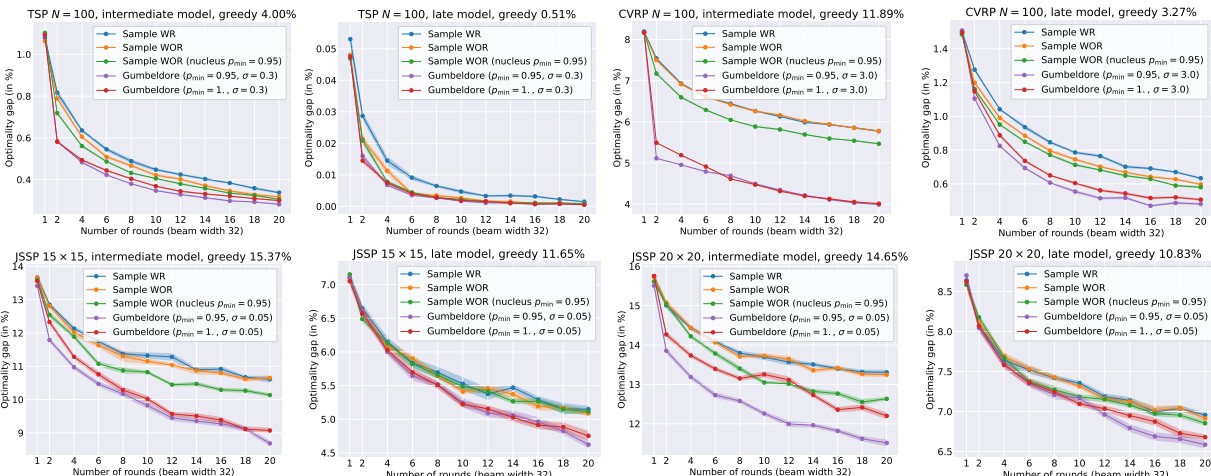

Figure 3: Sampling results for different training checkpoints, with the corresponding greedy performance of the model. For Sample WR, we sample $32 \cdot n$ sequences with replacement, where $n$ is the number of rounds. For TSP and CVRP, each data point corresponds to the average best solution across 100 instances. For JSSP, we evaluate the Taillard instances of the corresponding size. Sampling for each data point is repeated 20 times; shades denote standard errors.

also show the results for sampling without replacement with the growing nucleus, but without the policy update (6) ('Sample WOR (nucleus)'). We note that applying only one round of Sample WOR (nucleus) or GD is identical to Sample WOR by design. We use the same value for $\sigma$ as during training, and show results for $p_{\min} = 1$ (no nucleus sampling) and $p_{\min} = 0.95$ (as during training).

In line with Shi et al. (2020), we observe that sampling WOR yields a consistent but slight improvement over WR for the routing problems but is almost unobservable for JSSP, which has a much longer sequence length (225 for $15 \times 15$ and 400 for $20 \times 20$). Generally, allowing different nucleus sizes can significantly improve the intermediate models while giving enough room for exploration. Updating the policy between rounds with GD has a strong impact, especially on the intermediate models. Coupling the policy update with a growing nucleus further improves the performance. For example, with only four rounds, we obtain an absolute improvement over Sample WOR of approx. 2% for CVRP and >1% for JSSP $20 \times 20$ and reach an optimality gap with only 128 samples, which Sample WOR does not reach with 640 samples. While generally shrinking for only a few rounds, the relative improvement of GD is still significant in late models. In particular, GD consistently finds better solutions in late models when nucleus sampling only has limited influence. In Appendix C.4, we further illustrate the applicability of our method on a toy problem: we adapt our method for the board game Gomoku (Five in a Row) and evaluate how long it takes to consistently beat a deterministic expert bot.

## 6 Limitations and Future Work

Adjusting the policy by the advantage requires choosing a step size $\sigma$, which must be tuned for the problem class. To keep the approach principled, we left the step size fixed throughout training; however, the advantages' magnitude usually shrinks as training progresses. Hence, it might be advantageous to change $\sigma$ during training based on problem specifics or normalize the advantages as mentioned in Section 4.2.1. Our method requires keeping a search tree in memory instead of only one round of SBS, where we can discard all previous sequences after each step. However, keeping transitions in memory has benefits, such as only evaluating the policy when needed and thus reducing computation time during search.

## 7 Conclusion

In this work, we introduced an approach for training Neural Combinatorial Optimization models that bridges supervised learning and reinforcement learning challenges. During training, GD finds good solutions with only a few samples, eliminating the need for expert annotations while reducing computational demands. This method simplifies the training process and shows promise for enhancing the efficiency of Neural Combinatorial Optimization using larger architectures.

### Acknowledgments

We thank the Action Editor and the anonymous reviewers for their constructive feedback. This work was funded by the Deutsche Forschungsgemeinschaft (DFG, German Research Foundation) - 466387255 - within the Priority Programme "SPP 2331: Machine Learning in Chemical Engineering". The authors gratefully acknowledge the Competence Center for Digital Agriculture (KoDA) at the University of Applied Sciences Weihenstephan-Triesdorf for providing computational resources.

### Code Availability

Code for the experiments, data, and trained network weights are available at https://github.com/grimmlab/gumbeldore.

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

# A Background on Stochastic Beam Search

This section briefly describes the Gumbel-Top-k trick and the essence of *Stochastic Beams and Where to Find Them* (Kool et al., 2019c).

## A.1 The Gumbel-Top-k Trick

We denote by Gumbel($\mu$) a Gumbel distribution with *location* $\mu \in \mathbb{R}$ and unit scale. The distribution Gumbel(0) is the *standard* Gumbel distribution. Similarly to Kool et al. (2019c); Huijben et al. (2022), we write $G_\mu$ for a random variable following Gumbel($\mu$), and omit $\mu$ in the subscript in the case of standard location $\mu = 0$. The Gumbel distribution is closed under scaling and shifting, in particular for a standard $G \sim \text{Gumbel}(0)$ we have $\mu + G \sim \text{Gumbel}(\mu)$.

**Gumbel-Max and Gumbel-Top-k trick** The following is a condensed summary of the preliminaries in Kool et al. (2019c), with attribution to Gumbel (1954); Yellott Jr. (1977); Maddison et al. (2014); Vieira (2014). Consider a discrete distribution Categorical($p_1, \ldots, p_K$) with $K$ categories and the probability $p_i$ of the $i$-th category. Let $\phi_i$ be unnormalized log-probabilities (logits) for $p_i$, i.e., $p_i \propto \exp \phi_i$. We can obtain a sample from this distribution by *perturbing* each logit with a standard Gumbel and choosing the largest element ('Gumbel-Max trick'): Let $G_{\phi_i} = \phi_i + G_i$ with $G_i \sim \text{Gumbel}(0)$, then $G_{\phi_i} \sim \text{Gumbel}(\phi_i)$ by the shifting property and $\arg\max_i G_{\phi_i} \sim \text{Categorical}(p_1, \ldots, p_K)$.

In general, for any subset $B \subseteq \{1, \ldots, K\}$ we have

$$\max_{i \in B} G_{\phi_i} \sim \text{Gumbel}\left(\log \sum_{j \in B} \exp\left(\phi_j\right)\right), \text{ and} \tag{7}$$

$$\arg\max_{i \in B} G_{\phi_i} \sim \text{Categorical}\left(\frac{\exp\left(\phi_i\right)}{\sum_{j \in B} \exp\left(\phi_j\right)}, i \in B\right), \tag{8}$$

with max and arg max being independent.

The Gumbel-Max trick can be generalized to drawing an ordered sample $(i_1^*, \ldots, i_k^*)$ *without replacement* of size $k$ ('Gumbel-Top-$k$ trick') by finding the indices of the top $k$ perturbed logits (denoted by $\arg\text{top}\,k$):

$$i_1^*, \ldots, i_k^* = \arg\text{top}_i\,k\; G_{\phi_i}.$$

## A.2 Drawing Samples Without Replacement from a Sequence Model

**Beam search** Beam search is a low-memory, breadth-first tree search of limited *beam width $k$*. Given a policy $\pi$, the initial beam of a search from a root node (i.e., an empty sequence) with beam width $k$ consists of the top $k$ first tokens, ranked by their log-probability. Iteratively, at step $t$, each partial sequence in the beam expands its $k$ most probable tokens. The resulting expanded beam is again pruned down to $k$ by taking the top $k$ partial sequences according to their total log-probabilities $\log \pi(\boldsymbol{a}_{1:t}) = \sum_{t'=1}^{t} \log \pi(a_{t'} | \boldsymbol{a}_{1:t'-1})$, where $\boldsymbol{a}_{1:t}$ is any sequence in the expanded beam. Depending on the underlying factorized model, the final sequences found by beam search are often generic and lack variability (Vijayakumar et al., 2018).

**Stochastic Beam Search** In *Stochastic Beams and Where to Find Them* (Kool et al., 2019c), Kool et al. apply the Gumbel-Top-k trick in an elegant way to sample *sequences* without replacement from a sequence model. Assuming that the full true is instantiated with *all* possible complete (leaf) sequences $\boldsymbol{a}_{1:T}^{(i)}$ with $i \in \{1, \ldots, N\}$, one would sample distinct sequences with the Gumbel-Top-k trick by simply considering the perturbed log probabilities $G_{\phi_i} \sim \text{Gumbel}(\phi_i)$, where $\phi_i = \log \pi(\boldsymbol{a}_{1:T}^{(i)})$. Of course, instantiating the full tree is computationally infeasible in practice. Instead, Kool et al. make the following crucial observation: Identify a node in the trie by the set $S$ of leaves in its corresponding subtree and denote the corresponding (partial)

sequence by $\boldsymbol{a}^S$, with $\phi_S = \log \pi(\boldsymbol{a}^S)$. Then

$$\max_{i \in S} G_{\phi_i} \overset{(7)}{\sim} \mathrm{Gumbel}\left(\log \sum_{j \in S} \exp \phi_j\right) = \mathrm{Gumbel}\left(\sum_{j \in S} \pi(\boldsymbol{a}_{1:T}^{(j)})\right) = \mathrm{Gumbel}\left(\phi_S\right), \tag{9}$$

i.e., the maximum of the perturbed log probabilities of the leaf nodes in the subtree of $S$ follows a Gumbel distribution with location $\log \pi(\boldsymbol{a}^S)$. Thus, instead of instantiating the full tree to sample $k$ sequences, we can equivalently sample top-down (Maddison et al., 2014) and perform a beam search of width $k$ over the *perturbed* log-probabilities by recursively propagating the Gumbel noise down the subtree according to (9): Let $\boldsymbol{a}^S$ be any partial sequence in the beam at any step of the beam search with perturbed log-probability $G_{\phi_S}$ (which we can set to zero for the root node). Then we sample $G_{\phi_{S'}}$ for all direct children $S' \in \mathrm{Children}(S)$ *under the condition* that $\max_{S' \in \mathrm{Children}(S)} G_{\phi_{S'}} = G_{\phi_S}$, and use $G_{\phi_{S'}}$ as the scores for the beam search. We refer to the original paper by Kool et al. (2019c) for how to sample a set of Gumbels with a certain maximum.

Kool et al. (2019c) show on various language models that this simple procedure yields more diverse beam search results without sacrificing quality. From a sampling perspective, SBS can be seen as a principled way to randomize a beam search and construct stable unbiased estimators from a small number of sequences (Kool et al., 2019a; 2020).

## B  Policy Improvement Operation

We give a proof for the policy improvement obtained by Equation (1). The following lemma is part of the policy improvement proof for the updated action selection of Gumbel AlphaZero based on $Q$-value completion (Danihelka et al., 2022). For coherence, we formulate and prove it in our general setting, with attribution to Danihelka et al. (2022).

**Lemma:**  Consider a categorical distribution over the domain $\Omega = \{1, \ldots, n\}$ with corresponding probabilities $\pi(i)$ for $i \in \Omega$. Consider a map $q \colon \Omega \to \mathbb{R}$, and let $j \in \Omega$. Let $\pi'$ be the distribution obtained from $\pi$ by changing the unnormalized log-probability ('logit') of $j$ to

$$\mathrm{logit}\,\pi'(j) := \log \pi(j) + q(j) - \mathbb{E}_{i \sim \pi}\left[q(i)\right]. \tag{10}$$

Then, $\mathbb{E}_{i \sim \pi'}\left[q(i)\right] \geq \mathbb{E}_{i \sim \pi}\left[q(i)\right]$.

**Proof:**  We need to show that

$$\sum_{i \in \Omega} \pi'(i) q(i) \geq \sum_{i \in \Omega} \pi(i) q(i). \tag{11}$$

Let $y = q(j) - \mathbb{E}_{i \sim \pi}\left[q(i)\right]$. If $\pi(j) = 1$, then $y = 0$, so $\pi' = \pi$ and the claim is true. Hence, assume that $\pi(j) < 1$. Note that for any $i \in \Omega \setminus \{j\}$, we have $\pi'(i) = c \cdot \pi(i)$ with constant $c = \left(\sum_{i \in \Omega \setminus \{j\}} \pi(i) + \exp(y)\pi(j)\right)^{-1}$.

As $\sum_{i \in \Omega \setminus \{j\}} \pi(i) = 1 - \pi(j)$, we can rewrite $\mathbb{E}_{i \sim \pi}\left[q(i)\right]$ as

$$\mathbb{E}_{i \sim \pi}\left[q(i)\right] = \pi(j) q(j) + \sum_{i \in \Omega \setminus \{j\}} \pi(i) q(i) = \pi(j) q(j) + (1 - \pi(j)) \underbrace{\sum_{i \in \Omega \setminus \{j\}} \frac{\pi(i) q(i)}{\sum_{k \in \Omega \setminus \{j\}} \pi(k) q(k)}}_{=:\tilde{q}} \tag{12}$$

$$= \pi(j) q(j) + (1 - \pi(j)) \tilde{q} \tag{13}$$

$$= \pi(j)(q(j) - \tilde{q}) + \tilde{q}. \tag{14}$$

Analogously, we get

$$\mathbb{E}_{i \sim \pi'}\left[q(i)\right] = \pi'(j) q(j) + (1 - \pi'(j))(1 - \pi(j)) \sum_{i \in \Omega \setminus \{j\}} \frac{c \pi(i) q(i)}{\sum_{k \in \Omega \setminus \{j\}} c \pi(k) q(k)} \tag{15}$$

$$= \pi'(j)(q(j) - \tilde{q}) + \tilde{q}, \tag{16}$$

as the constant $c$ cancels out. In particular, equivalently to (11), we can show that

$$\pi'(j)(q(j) - \tilde{q}) \geq \pi(j)(q(j) - \tilde{q}). \tag{17}$$

By the policy update, we have $\pi'(j) > \pi(j) \iff y > 0$, so (17) follows if we can show that $y > 0 \iff (q(j) - \tilde{q}) > 0$. But this is true, as

$$y > 0 \iff q(j) > \mathbb{E}_{i \sim \pi}[q(i)] \iff q(j) > \pi(j)q(j) + (1 - \pi(j))\tilde{q} \tag{18}$$
$$\iff (1 - \pi(j))q(j) > (1 - \pi(j))\tilde{q} \tag{19}$$
$$\overset{\pi(j) < 1}{\iff} q(j) > \tilde{q} \tag{20}$$
$$\iff q(j) - \tilde{q} > 0. \tag{21}$$

$\square$

We can now prove the policy improvement over the entire sequence model.

**Proposition:** Let $\pi$ be a policy, $\sigma > 0$ and let $\boldsymbol{a}_{1:T} = (a_1, \ldots, a_T)$ be a full sequence drawn from $\pi$. Let $\pi'$ be the policy obtained from $\pi$ by changing the logit of $\pi'(a_i | \boldsymbol{a}_{1:i-1})$ for all $i \in \{1, \ldots, T\}$ to

$$\text{logit } \pi'(a_i | \boldsymbol{a}_{1:i-1}) := \log \pi(a_i | \boldsymbol{a}_{1:i-1}) + \\ \sigma \cdot \left( \mathbb{E}_{\boldsymbol{a}'_{1:T} \sim \pi(\cdot | \boldsymbol{a}_{1:i})}[f_x(\boldsymbol{a}'_{1:T})] - \mathbb{E}_{\boldsymbol{a}'_{1:T} \sim \pi(\cdot | \boldsymbol{a}_{1:i-1})}[f_x(\boldsymbol{a}'_{1:T})] \right). \tag{22}$$

Then,

$$\mathbb{E}_{\boldsymbol{a}_{1:T} \sim \pi'}[f_x(\boldsymbol{a}_{1:T})] \geq \mathbb{E}_{\boldsymbol{a}_{1:T} \sim \pi}[f_x(\boldsymbol{a}_{1:T})]. \tag{23}$$

**Proof:** We omit $x$ in the subscript of $f$ to simplify the notation. As $\sigma$ only changes the magnitude of the step, we can assume without loss of generality that $\sigma = 1$. We show the claim by induction on the length $T$.

For $T = 1$, we have $\boldsymbol{a}_{1:T} = (a_1)$, and the update reduces to

$$\text{logit } \pi'(a_1) = \log \pi(b) + f(a_1) - \mathbb{E}_{a' \sim \pi}[f(a')]. \tag{24}$$

Hence, the claim follows from the lemma above.

For the induction step for arbitrary $T$, we have

$$\mathbb{E}_{\boldsymbol{a}'_{1:T} \sim \pi'}[f(\boldsymbol{a}_{1:T})] = \sum_{\boldsymbol{a}'_{1:T}} \pi'(\boldsymbol{a}'_{1:T}) f(\boldsymbol{a}'_{1:T}) \tag{25}$$

$$= \sum_{a'_1} \pi'(a'_1) \mathbb{E}_{\boldsymbol{a}'_{1:T} \sim \pi'(\cdot | a'_1)}[f(\boldsymbol{a}'_{1:T})] \tag{26}$$

$$\geq \sum_{a'_1} \pi'(a'_1) \mathbb{E}_{\boldsymbol{a}'_{1:T} \sim \pi(\cdot | a'_1)}[f(\boldsymbol{a}'_{1:T})], \tag{27}$$

where the last inequality follows from the induction hypothesis. But then, due to the form of the policy update (22), the lemma above can be applied again and we get

$$\sum_{a'_1} \pi'(a'_1) \mathbb{E}_{\boldsymbol{a}'_{1:T} \sim \pi(\cdot | a'_1)}[f(\boldsymbol{a}'_{1:T})] \geq \sum_{a'_1} \pi(a'_1) \mathbb{E}_{\boldsymbol{a}'_{1:T} \sim \pi(\cdot | a'_1)}[f(\boldsymbol{a}'_{1:T})] \tag{28}$$

$$= \sum_{\boldsymbol{a}'_{1:T}} \pi(\boldsymbol{a}'_{1:T}) f(\boldsymbol{a}'_{1:T}) \tag{29}$$

$$= \mathbb{E}_{\boldsymbol{a}'_{1:T} \sim \pi}[f(\boldsymbol{a}'_{1:T})], \tag{30}$$

what we wanted to show. $\square$

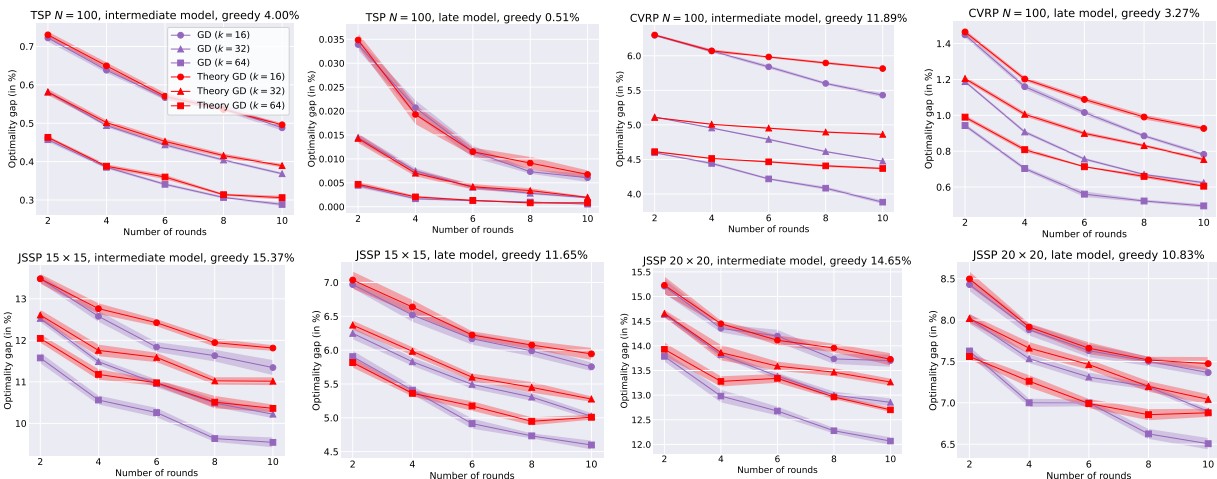

Figure 4: Comparison of the practical policy update (6) 'GD' with the theoretical update (1) 'Theory GD' across various model checkpoints and beam widths $k \in \{16, 32, 64\}$. For TSP and CVRP, each data point corresponds to the average best solution across 100 instances. For JSSP, we evaluate the Taillard instances of the corresponding size. Nucleus sampling is switched off. Sampling for each data point is repeated 10 times; shades denote standard errors.

## C   Extended Experimental Results

### C.1   Comparison with Theoretical Policy Improvement

We compare the practical policy update ('GD') that we use in our method (6) with the policy update (1) that yields a theoretical policy improvement ('Theory GD'). The difference between them is that GD updates the logits in a trajectory according to a single 'global' advantage. This global advantage is the difference between the outcome of the trajectory and the estimated expected outcome of the full policy, and it is propagated recursively to all ancestors of the corresponding leaf node. In contrast, Theory GD updates the logit of each ancestor individually by computing the difference 'locally' between the expected outcome of a node and its direct parent. While Theory GD yields a provable policy improvement, there are two practical difficulties: i.) In GD, we only need to compute a single expected value. For Theory GD, we must compute the expected value for each encountered node individually. ii.) In GD, we can use all $k$ sampled trajectories to compute the expected outcome of the full policy. For Theory GD, when computing the advantage of a particular node, we need to estimate the expectation of the node and its parent. In particular, we can only get a non-zero advantage if there are at least two trajectories through the parent. Depending on the confidence of the model and the beam width $k$, this is often not the case for nodes deeper in the tree.

In Figure 4 we compare Theory GD with our chosen practical update GD over different checkpoints and beam width $k \in \{16, 32, 64\}$ as in Section 5.3. We use the same values for $\sigma$ as in training, but do not apply nucleus sampling. GD and Theory GD perform similarly on the TSP, where the model is confident early on. While Theory GD is still a strong update method on the CVRP and JSSP, GD outperforms it in almost all cases. Coupled with the fact that GD only needs to compute a single expected value and is easier to implement, we consider it more advantageous for neural CO. However, in many cases, the results are close, which illustrates the theoretical rationale for GD.

### C.2   Comparison with Self-Improving Training Method for LEHD on TSP

Concurrently to Drakulic et al. (2023), Luo et al. (2023) propose the 'Light Encoder Heavy Decoder' (LEHD) model for routing problems and also show promising generalization capabilities. The structure of LEHD is similar to BQ; however, while BQ only shares an affine embedding of the nodes across all time steps, LEHD encodes the nodes with a single attention layer. The decoder consists of six attention layers, and as in BQ,

Table 3: Optimality gaps for TSP compared with LEHD with and without labeled data. For LEHD RL+RRC+SL, we take the results from the original paper, which are better than the ones we obtained with our implementation.

| Method | Test | Generalization | | |
|---|---|---|---|---|
| | TSP $N = 100$ | TSP $N = 200$ | TSP $N = 500$ | TSP $N = 1000$ |
| LEHD SL, greedy | 0.58% | 0.95% | 1.72% | 3.34% |
| LEHD RL+RRC+SL, greedy | 1.07% | 1.45% | 2.56% | 4.52% |
| **LEHD SI GD (ours)**, greedy | 0.40% | 0.72% | 1.43% | 3.30% |

Table 4: Inference with Gumbeldore on the CVRP. For $N = 100$, we only consider a test subset of 1000 instances. SBS and GD results are averaged over five repetitions. We use nucleus sampling with constant $p = 0.8$ for SBS and GD, and keep $\sigma = 3.0$ for GD.

| Method | Test | Generalization | | |
|---|---|---|---|---|
| | $N = 100$ Gap | $N = 200$ Gap | $N = 500$ Gap | $N = 1000$ Gap |
| Beam 128 | 1.16% | 1.10% | 1.72% | 4.28% |
| Beam 256 | 1.00% | 0.97% | 1.60% | 4.27% |
| Beam 512 | 0.86% | 0.90% | 1.40% | 3.85% |
| SBS $32 \times 4$ | 0.80% | 1.07% | 2.44% | 6.04% |
| SBS $32 \times 8$ | 0.65% | 0.87% | 2.34% | 6.00% |
| SBS $64 \times 8$ | 0.51% | 0.73% | 2.20% | 5.80% |
| GD $32 \times 4$ | 0.73% | 1.03% | 2.13% | 5.25% |
| GD $32 \times 8$ | 0.59% | 0.76% | 2.03% | 5.12% |
| GD $64 \times 8$ | 0.49% | 0.56% | 1.88% | 4.76% |

the model is trained to predict the next node of random subtours sampled from expert solutions. Luo et al. (2023) additionally present a training process on TSP that does not require solutions from solvers as follows:

- Train LEHD on TSP $N = 20$ with self-critical REINFORCE for some time.

- Generate a training set of 200,000 random instances of TSP $N = 100$ and generate solutions for them with the current model.

- Sample subtours for each instance and randomly reconstruct (RRC) them to improve the solutions.

- Continue training of LEHD by supervised learning with the generated dataset.

In Table 3, we compare this approach (LEHD RL+RRC+SL) to their supervised approach and our method. We see that our simple training process based on sampling surpasses the relatively complicated RL + randomly reconstructed solutions approach and obtains results comparable to its supervised counterpart.

### C.3 Gumbeldore at Inference Time

There is a plethora of work on how to exploit an already trained policy for neural CO, such as Simulation-Guided Beam Search (Choo et al., 2022) or (Efficient) Active Search (Bello et al., 2016; Hottung et al., 2022) (cf. Section 2). Although we consider our sampling method mainly for training, in this section, we evaluate the performance during inference in the low sample regime. High exploration plays a subordinate role in this setting, so we use Top-$p$ sampling with a *constant $p$* in all rounds. Table 4 shows inference results using the model trained with SI GD. We compare GD $k \times n$ with $n$ rounds and beam width $k$ with sampling without

Table 5: Inference with Gumbeldore on the JSSP. Results are averaged over five repetitions. In all cases, we use nucleus sampling with constant $p = 0.8$, and keep $\sigma = 0.05$ for GD.

| Method | $15 \times 15$ Gap | $20 \times 15$ Gap | $20 \times 20$ Gap | $30 \times 15$ Gap | $30 \times 20$ Gap | $50 \times 15$ Gap | $50 \times 20$ Gap | $100 \times 20$ Gap |
|---|---|---|---|---|---|---|---|---|
| SBS $32 \times 4$ | 5.0% | 5.7% | 7.3% | 6.1% | 9.4% | 1.1% | 4.3% | 0.4% |
| SBS $32 \times 8$ | 4.7% | 5.3% | 7.0% | 5.7% | 9.0% | 0.9% | 4.0% | 0.3% |
| SBS $64 \times 8$ | 4.3% | 5.3% | 6.4% | 5.2% | 8.8% | 0.9% | 4.0% | 0.3% |
| GD $32 \times 4$ | 4.8% | 5.6% | 7.0% | 5.8% | 9.2% | 1.1% | 4.3% | 0.4% |
| GD $32 \times 8$ | 4.5% | 5.3% | 6.7% | 5.3% | 8.8% | 0.9% | 4.0% | 0.3% |
| GD $64 \times 8$ | 4.2% | 4.8% | 6.3% | 5.0% | 8.6% | 0.8% | 3.7% | 0.2% |

replacement (with Top-$p$) via round-based SBS $k \times n$ and to beam search with corresponding beam width. We see that both sampling methods outperform beam search on $N = 100$ and $N = 200$ but not on $N = 500$ and $N = 1000$. Furthermore, the policy update of GD leads to a consistent improvement over SBS. The CVRP results indicate that combining GD and deterministic beam search could result in a robust inference method. On the JSSP in Table 5, we observe the same improvement of GD over SBS. The margin, however, is not as wide as for intermediate models by Section 5.3. We omit the results for deterministic beam search on JSSP, as SBS supersedes them with nucleus sampling and a beam width of 16 (cf. Table 2).

### C.4   Gomoku: A Toy Problem

Played on a Go board, Gomoku is a game for two players who take turns placing their stones on the board, starting with black. The first player to have five stones in a row horizontally, vertically, or diagonally wins. We formulate the following problem: Given a deterministic expert bot playing black, how long does it take a greedy model playing white to learn from scratch to beat the opponent from all possible starting positions of black? Here, the model's policy takes a board configuration and predicts a distribution over the next possible moves. We take the deterministic rule-based bot ('v1'), the Gomoku state representation, and the AlphaZero architecture for Go provided by the LightZero benchmark suite (Niu et al., 2023). In each epoch, we generate training data by sampling multiple trajectories using the current policy from all possible starting positions of black. Then, for each start position, we take from the sampled trajectories either a random trajectory where white wins or, if there is no win for white, a random trajectory where black wins. In the case of a draw, we take a random draw. The supervised training in each epoch then consists of learning the policy to predict the next move of the winner. After each epoch, we greedily unroll the policy for each starting position of black and count the number of times white wins. *How many epochs does it take for white to win every time?* The formulated problem is not standard, and board games are usually approached with self-play. Therefore, it is instead a tiny illustrative toy problem to further show the applicability of our method.

We play on a $9 \times 9$ Go board. First, we let the model sample 320 sequences with replacement for each possible starting position and train the policy in each epoch on 100 batches of 32 pairs of the form (<board configuration>, <next move of winner>). Repeated ten times, the model takes an average of **60.2 epochs** for white to consistently beat black.

For training with GD, we score a complete trajectory with its game result: 1 for a white win, -1 for a black win, and 0 for a draw. We also sample 320 sequences in 10 rounds with an SBS batch size of 32. We completely turn off nucleus sampling and use the game outcome as an advantage to propagate up the tree (equivalent to setting the expected outcome to a draw). Using a step size of $\sigma = 5$, which heavily favors moves that lead to a win and punishing moves that lead to a lose, it takes an average of **47.7 epochs** ($\sim 20\%$ faster than sampling with replacement) for white to greedily win all the time.

# D  Implementation and Experimental Details

**Hardware and frameworks**  Our code is developed in PyTorch (Paszke et al., 2019). Training and inference are performed for all experiments using two NVIDIA RTX A5000 with 24GB memory. As we do not need to collect gradients during sampling, we can easily parallelize solution sampling for different instances in each epoch. We spread it across 16 workers using ray.io.

## D.1  Choice of Gumbeldore Step Size

We generally choose the step size $\sigma$ for the update in (6) as follows: We train the model using round-based SBS (i.e., with $\sigma = 0$) for 20 epochs with four rounds of beam width 32. We then perform a grid search over $\sigma \in [0, 5]$ on a small validation set.

## D.2  Traveling Salesman Problem

### D.2.1  Problem Setup

In the Euclidean TSP with $N$ nodes in the unit square $[0, 1]^2 \subseteq \mathbb{R}^2$, a node permutation (i.e., a complete roundtrip, where all nodes are visited only once) with minimal edge weight should be found. A problem instance is defined by the two-dimensional coordinates of the $N$ nodes. The sequential problem asks for choosing one unvisited node at a time. The objective function to maximize evaluates a full tour by the negative tour length. Random instances are generated in the standard way of Kool et al. (2019b) by sampling the coordinates of the $N$ nodes uniformly from the unit square.

### D.2.2  Policy Network

We use the transformer-based architecture of BQ (Drakulic et al., 2023) and give a short overview of the network flow: At the start, the coordinates of the $N$ nodes $(\boldsymbol{x}_1, \ldots, \boldsymbol{x}_N) \in \mathbb{R}^{N \times 2}$ are affinely embedded into a latent space of dimension $d$, and we obtain nodes $(\boldsymbol{x}'_1, \ldots, \boldsymbol{x}'_N) \in \mathbb{R}^{N \times d}$. At each time step, given a partial tour with an origin node $\boldsymbol{x}'_i$ and a destination node $\boldsymbol{x}'_j$, we add a learnable lookup embedding to $\boldsymbol{x}'_i$ and $\boldsymbol{x}'_j$ and send them together with all unvisited nodes through a stack of transformer layers. ReZero (Bachlechner et al., 2021) normalization is used instead of layer normalization within the transformer layers. No positional encoding is used, as the order of nodes does not play a role. After the transformer layers, a linear layer $\mathbb{R}^d \rightarrow \mathbb{R}$ projects the processed nodes to a logit vector in $\mathbb{R}^N$, from which the logits corresponding to $\boldsymbol{x}'_i$ and $\boldsymbol{x}'_j$ are masked out.

**Trajectory prediction**  Given an instance of $N$ nodes, we choose a random node as the origin, mark it as visited, and set it as the destination node as well. The network predicts a distribution over the remaining nodes, and after choosing one, we mark this one as visited, set it as the new origin node, and so on.

We use a latent dimension of $d = 128$ (following a preprint version of BQ) and nine transformer layers with eight heads and feed-forward dimension of 512.

### D.2.3  Supervised Training

**Subtours**  Given an instance and a complete tour, we train on random *subtours*. That is, given $t \in \{4, \ldots, N\}$ (a subtour with three nodes is trivial) and an instance of size $N = 100$ with a complete tour, we sample a subtour of length $t$. The first and last nodes of the subtour are taken as the origin and destination nodes, and the corresponding target to predict is the next node after the origin node. For computational efficiency, we keep $t$ fixed within the same minibatch.

**Subtour augmentation**  During training, we randomly augment each sampled subtour in the following ways, each of which does not change the solution:

- Switch the direction of the subtour

- Swap x and y coordinates for each node

- Reflect all nodes along the horizontal line going through the center $(0.5, 0.5)$

- Reflect all nodes along the vertical line going through the center $(0.5, 0.5)$

- Rotate all nodes around the center $(0.5, 0.5)$ by a random angle. Note that this can lead to coordinates outside $[0, 1]^2$. In this case, we linearly scale the coordinates so that they lie within the unit square again.

**Hyperparameters**  In each epoch, we train on 1,000 batches consisting of 1,024 sampled subtours. We use Adam (Kingma & Ba, 2014) as an optimizer, with an initial learning rate of 1e-4 and no decay. Gradients are clipped to unit $L_2$-norm.

### D.2.4  Gumbeldore Training

In each epoch, we sample 1,000 random instances for which we sample 128 solutions with GD using a beam width of $k = 32$ in $n = 4$ rounds. We use a step size of $\sigma = 0.3$ throughout training. We start with $p_{\min} = 1$ (i.e., no nucleus sampling) to not restrict exploration in any way, and set $p_{\min} = 0.95$ after 500 epochs. Supervised training is performed with the best sampled solutions as in its supervised counterpart; however, we use an initial learning rate of 2e-4. To better utilize the GPUs, we parallelize the sampling procedure across 16 workers which share the two GPUs. Sampling all solutions for the 1,000 instances takes about one minute for $N = 100$.

## D.3  Capacitated Vehicle Routing Problem

### D.3.1  Problem Setup

A CVRP problem instance is given by coordinates of $N$ customer nodes and one depot node. Each customer node $\boldsymbol{x}_i$ has a *demand* $\delta_i$, which must be fulfilled by a delivery vehicle of capacity $D$. The vehicle must visit all nodes exactly once in a set of subtours that start and end at the depot node, and where the sum of the customers' demands visited in a subtour does not exceed the vehicle's capacity $D$. The goal is to find a set of subtours with minimal total distance that visits all customers.

**Instance generation**  Following (Kool et al., 2019b; Drakulic et al., 2023; Luo et al., 2023), an instance is generated by sampling the coordinates for the customers and the depot uniformly from the unit square. The demands $\delta_i$ are sampled uniformly from the set $\{1, \ldots, 9\}$. The vehicle capacity is set respectively to $D = 50, 80, 100, 250$ for corresponding $N = 100, 200, 500, 1000$. We normalize the vehicle's total capacity to $\hat{D} = 1$ and the demands to $\hat{\delta}_i = \frac{\delta_i}{D}$.

**Solution formation**  To align solutions, we also follow the approach of (Kool et al., 2019b; Drakulic et al., 2023; Luo et al., 2023) and describe a complete solution by *two* vectors, where one is a permutation of the customer indices, and the other is a binary vector indicating whether the $i$-th customer in the permutation is reached via the depot or not. For example, a complete tour $(0, 1, 4, 5, 0, 2, 3, 0, 6, 7, 8, 0)$, where index 0 denotes the depot, consists of 3 subtours which start and end at the depot. The corresponding solution split into the two vectors of length 8 is $(1, 4, 5, 2, 3, 6, 7, 8)$ for the permutation and $(1, 0, 0, 1, 0, 1, 0, 0)$.

### D.3.2  Policy Network

As for the TSP, we use the transformer-based architecture of BQ (Drakulic et al., 2023) and give a short overview of the network flow: At any point in time, we represent each node (including the depot) as a four-dimensional vector, where the first two entries are the coordinates of the node, the third entry is the demand (with 0 demand for depot) and the last entry is the current remaining capacity of the vehicle. Also, at any point, we have an origin node (equal to the depot at the beginning). The depot, the origin node, and the remaining unvisited nodes are affinely embedded into the latent space $\mathbb{R}^d$, and we mark the origin and the depot by adding a learnable lookup embedding as for the TSP. The stack of transformer layers processes

the embeddings of the depot, origin, and remaining nodes before projecting the output to two logits for each remaining node via a linear layer. The two logits correspond to choosing the node via the depot or directly from the origin. We mask all infeasible actions, indicating whether a node must be reached via the depot as its demand exceeds the current capacity of the vehicle (or must be reached via the depot as the origin is equal to the depot). The structure of the attention layers is identical to the ones in the TSP.

As in the original paper (Drakulic et al., 2023), we use a latent dimension of $d = 192$, nine transformer layers with 12 heads, and a feed-forward dimension of 512.

### D.3.3 Supervised Training

**Subtours** As for the TSP, we train on random subtours. We impose the restriction that a sampled subtour *must* end at the depot (Luo et al., 2023).

**Subtour augmentation** We augment an instance by randomly reversing the direction of individual subtours. We then *sort* the subtours in ascending order by the remaining vehicle capacity at the end of the subtour. Sorting the subtours is a crucial step that we copy from Drakulic et al. (2023), who analyze that the order of the subtours has a substantial impact on the final performance of the model, where the model obtains better results when learning to schedule subtours first which utilize the vehicle as good as possible. Finally, we perform the same random geometric augmentation techniques as for TSP (reflection, rotation, flipping).

**Inference** During inference, we follow the approach of Drakulic et al. (2023) and consider at each step the 250 nearest neighbors of the current origin node.

**Hyperparameters** The hyperparameters for training are identical to TSP.

### D.3.4 Gumbeldore Training

Gumbeldore hyperparameters are identical to TSP, with a step size of $\sigma = 3.0$. To enhance generalization, we sample the vehicle capacity from $\{40, 41, \ldots, 100\}$ during instance generation.

## D.4 Job Shop Scheduling Problem

### D.4.1 Problem Setup

**Problem definition** We use a similar notation as in Pirnay et al. (2023). In a JSSP instance of size $J \times M$, we are given $J$ jobs. Each job consists of $M$ operations which need to be scheduled on $M$ machines. There is a bijection between the operations of a job and the set of the machines, i.e., every job must visit each machine exactly once. Thus, job $i \in \{1, \ldots, J\}$ can be represented by $(o_{i,l}, p_{i,l})_{l=1}^{M}$, where $o_{i,l} \in \{1, \ldots, M\}$ is the index of the machine on which the $l$-th operation must run, and $p_{i,l} \in \mathbb{R}_{>0}$ is the *processing time* that it takes to process the operation on machine $o_{i,l}$. The operations of a job must run in order; only one operation can be processed by a machine at a time, and once an operation starts, it must finish. The aim is to find a *schedule* with minimum makespan, where the makespan is given by the time the last machine finishes. In particular, the objective function to maximize is defined by the negative makespan of a schedule. A JSSP instance is fully defined by the set $\{(o_{i,l}, p_{i,l})_{l=1}^{M}\}_{i=1}^{J}$.

**Instance generation** We generate a random instance in the way of Taillard (1993) by uniformly sampling processing times from $\{1, \ldots, 99\}$ and setting the order of the machines on which a job must run as a uniformly random permutation of the set of machines.

**Schedule representation** As there is a bijection between the operations of a job and the $M$ machines, and the operations must run in order, we can represent a schedule by an *ordered sequence of job indices* $(j_1, \ldots, j_{J \cdot M})$, where $j_i \in \{1, \ldots, J\}$. Here, the occurrence of a job $i$ means that the next unscheduled operation of job $i$ should be processed on the corresponding machine as soon as possible. Note that, as in the routing problems, a sequence defining a schedule is generally not unique. In particular, in our constructive sequential formulation, we choose one unfinished job index after another until all jobs are finished.

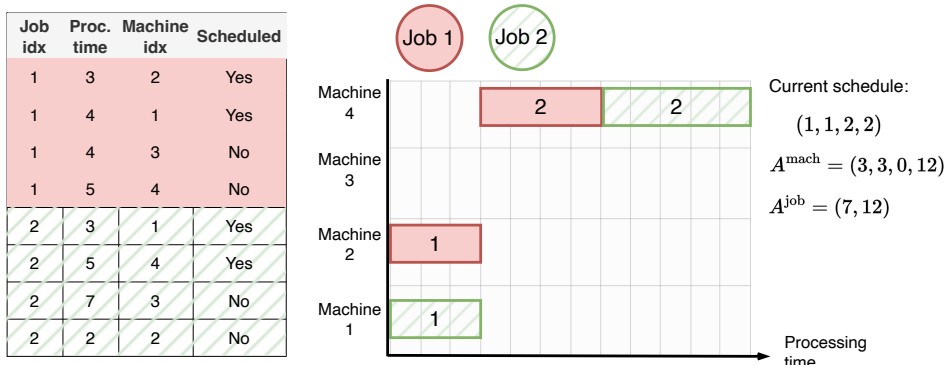

Figure 5: Example of a problem instance with $J = 2$ and $M = 4$. The operations' processing times and machines are shown in the table on the left. The first two operations have already been scheduled with the partial schedule $(1, 1, 2, 2)$. The corresponding Gantt chart and availability times are shown on the right.

**Tail subproblems**    Similar to the routing problems, we want to employ a similar tail-recursion property. For this, let $\{(o_{i,l}, p_{i,l})_{l=1}^{M}\}_{i=1}^{J}$ be a problem instance. We introduce two additional tuples $(A_i^{\mathrm{mach}})_{i=1}^{M}$ and $(A_i^{\mathrm{job}})_{i=1}^{J}$, where $A_i^{\mathrm{mach}}$ indicates the *availability* of the $i$-th machine, i.e., the earliest time an operation can start on machine $i$. Analogously, $A_i^{\mathrm{job}}$ indicates the earliest time the next operation of the $i$-th job can start. For an empty schedule, the availability times are all set to zero. When choosing a job with index $i$, let $l$ be the index of the operation that is to be scheduled and $m = o_{i,l}$ be the index of the machine on which the operation must run. We then compute the finishing time $z$ of the operation via

$$z = \max\{A_i^{\mathrm{job}}, A_m^{\mathrm{mach}}\} + p_{i,l} \tag{31}$$

and update the availability times to

$$A_m^{\mathrm{mach}}, A_i^{\mathrm{job}} \leftarrow z. \tag{32}$$

The crucial thing to note here is that given an optimal schedule $(j_1, \ldots, j_{J \cdot M})$ with minimal makespan and $q \in \{1, \ldots, J \cdot M\}$, the subschedule $(j_q, \ldots, j_{J \cdot M})$ is an optimal solution to the subproblem where after scheduling $j_1, \ldots, j_{q-1}$, the corresponding operations have been erased, and the availability times have been updated according to (32). A small example can be seen in Figure 5

This property motivates how our model is set up: For the routing problems, we have a heavy decoder that only cares about unvisited nodes and no longer about how it visited previous nodes. Similarly, for the JSSP, we are only interested in the unscheduled operations and the availability times at any time. We feed this information to a network described below.

### D.4.2   Policy Network

A suitable architecture for the JSSP should account for the fact that the problem is invariant to the job and the machine indexing (for the latter, it is only important if two operations must run on the same machine, but not which one). Our proposed architecture processes the $J \cdot M$ operations with multiple transformer layers, and we use masking to account for the indexing invariance. We make this precise as follows:

Let $A^{\mathrm{mach}}, A^{\mathrm{job}}$ be the availability times at any constructive step. Let $i$ be the index of an unfinished job, and $m_i$ be the index of the machine on which the next unscheduled operation of job $i$ must run. Define $r_i := \max\{A_i^{\mathrm{job}}, A_{m_i}^{\mathrm{mach}}\}$ as the time at which this operation would start if it were scheduled next. We represent the state of the instance by $\boldsymbol{A} \in \mathbb{R}^{J \times M \times 2}$, where for $i \in \{1, \ldots, J\}$ and $l \in \{1, \ldots, M\}$ we set

$$\boldsymbol{A}_{i,l} = \left( \frac{p_{i,l}}{100}, \frac{r_i - \min_{i' \in \{1, \ldots, J\}} r_{i'}}{100} \right). \tag{33}$$

I.e., we repeat the start time across all operations for a job, shift the time back to 0, and scale the processing times into $[0, 1]$. We affinely embed $\boldsymbol{A}$ into a latent space $\mathbb{R}^d$ to obtain $\hat{\boldsymbol{A}} \in \mathbb{R}^{J \times M \times d}$. We compute a sinusoidal positional encoding $\boldsymbol{P} \in \mathbb{R}^{M \times d}$ (Vaswani et al., 2017) which we add to each job $i$ via $\hat{\boldsymbol{A}}_i + \boldsymbol{P}$. We send the resulting sequence of operations through a stack of *pairs* of transformer layers, wherein each pair:

    i.) We use ReZero normalization (Bachlechner et al., 2021) as in the architecture for the routing problems.

    ii.) In the first transformer layer, referred to as the 'job-wise layer':

- Only operations within the same job attend to each other. In practice, this can be efficiently achieved by folding the job dimension into the batch dimension, i.e., reshaping the sequence from $\mathbb{R}^{B \times J \times M \times d}$ to $\mathbb{R}^{B \cdot J \times M \times d}$.
- We mask all operations that have already been scheduled.
- We add ALiBi (Press et al., 2022) positional information to query-key attention scores as an attention bias in each head of a job-wise layer. Precisely, let $h$ be the number of heads, then the ALiBi-specific *slope* for the $k$-th head is given by the $2^{-\frac{8k}{h}}$. Then, we set the additive bias for a querying operation with index $l_1 \in \{1, \dots, M\}$ and a key operation with index $l_2$ to $2^{-\frac{8k}{h}} \cdot (l_2 - l_1)$ for the $k$-th head. While not strictly necessary for good performance, we found that ALiBi slightly improves the results by carrying the positional information of the operations through all job-wise layers.

    iii.) In the second transformer layer, referred to as the 'machine-wise layer':

- We let two operations attend to each other only if they need to run on the same machine.
- We also mask all operations that have already been scheduled but do not employ ALiBi attention bias.

The operations transform according to different roles by repeatedly switching between job- and machine-wise layers. Note that this strategy is invariant to the job and machine indexing. In principle, it is possible to achieve a similar effect by only considering the entire sequence in $\mathbb{R}^{B \times J \cdot M \times d}$ for all transformer layers and using different masks within individual heads to account for job-wise and machine-wise attention. We settled for the switching strategy, as folding the jobs into the batch dimension practically saves computation time.

After the transformer layers, we gather the output $\boldsymbol{O} \in \mathbb{R}^{J \times d}$ corresponding to the next unscheduled operation of each job (in case the job is already finished, we take the last operation ). We apply a final transformer block on the sequence $\boldsymbol{O}$, masking already finished jobs. Afterward, we project the output to logits with a linear layer $\mathbb{R}^d \to \mathbb{R}$, again masking finished jobs.

Figure 6 gives an overview of the network.

**Size**    We use a latent dimension of $d = 64$ with three pairs of transformer layers, where each layer has eight heads and a feedforward dimension of 256. With the final transformer layer, this amounts to a total of 7 layers.

### D.4.3 Gumbeldore Training

We train the model with GD for 100 epochs. In each epoch, we randomly pick a $J \times M$ size in $\{10 \times 15, 15 \times 15, 15 \times 20\}$. We generate 512 random instances of the chosen size for which we sample 128 solutions with GD using a beam width of $k = 32$ in $n = 4$ rounds. We use a fixed advantage step size of $\sigma = 0.05$. We start with $p_{\min} = 1$ and set $p_{\min} = 0.95$ after 50 epochs. During supervised training on the generated solutions, given a full trajectory $(j_1, \dots, j_{J \cdot M})$, we uniformly sample $q \in \{1, \dots, J \cdot M - 1\}$ and compute $A^{\text{mach}}$ and $A^{\text{job}}$ according to the subschedule $(j_1, \dots, j_{q-1})$. The training target is then to predict $j_q$. No further augmentation is performed.

As for the routing problems, we use an initial learning rate of 2e-4 and clip gradients to unit norm.

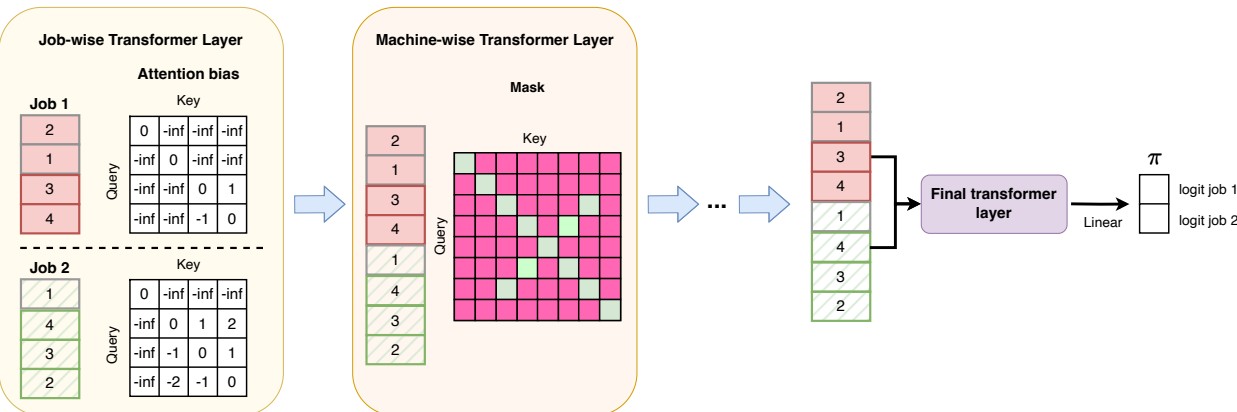

Figure 6: Overview of the network architecture, using the two jobs with four operations from Figure 5. Boxes represent operations, and the numbers in the boxes indicate the machine on which it has to run. Here, we assume that the first two operations of Job 1 and the first operation of Job 2 are already scheduled. In the job-wise transformer layer, only operations within the same job attend to each other. We mask scheduled operations. We accommodate masking and ALiBi positional information in the additive attention bias (shown before scaling with slope). In the machine-wise layer, only operations running on the same machine may attend to each other, also masking scheduled operations (pink square: mask, green square: do not mask). The next possible operation is gathered for each job and used to predict the policy.

# E    Comparison to Concurrent Work

Independently of our work, Corsini et al. (2024) develop a similar learning strategy for the JSSP. They also turn away from RL and sample 256 solutions in each epoch for randomly generated instances using the current model. Then, they supervisedly train a GNN + Pointer Network on the generated solutions. They call this strategy 'self-labeling'. While our Algorithm 1 is slightly different from theirs (maintaining a best greedy policy and expanding the dataset in case of no improvement), their work and ours match in the spirit of applying behavior cloning to sampled solutions. However, Corsini et al. (2024) focus on the JSSP and their proposed GNN architecture using the disjunctive graph representation of the JSSP. Most importantly, while they discuss different sampling schemes, they settled on sampling with replacement (Monte Carlo i.i.d. sampling) from the sequence model, which they found sufficient for their approach. Our main contribution revolves around a principled way to get the most out of a few samples, applied to various neural CO problems, and we show in Section 5.3 that our method can significantly improve the sampling performance. Furthermore, we compare our method on the JSSP with their approach in Table 2. Our work is also motivated by bridging the problem that large network architectures that generalize strongly can be computationally challenging to train with policy gradient methods. However, we emphasize that we believe the work of Corsini et al. (2024) and ours are mutually reinforcing, as both conclude that a more 'straightforward' training strategy than self-critical policy gradient methods can lead to a strong performance in neural CO.

