# OpenReview forum: "Self-Improvement for Neural Combinatorial Optimization: Sample Without Replacement, but Improvement"
_TMLR — Accepted by TMLR_

### Review · Reviewer_WWT5 · 2024-05-01

**Summary Of Contributions:**

The paper proposes a new method for neural combinatorial optimization that trains a neural network policy based on samples from a sampling method proposed in the paper. The paper first introduces combinatorial optimization (CO), as well as past machine learning approaches for solving CO. The paper argues that past reinforcement learning based approaches can be challenging to train and then outline their approach for training a policy based on trajectories sampled as pseudo-expert trajectories. At the end of the introduction, the authors outline their contributions focusing on the approach, the sampling method and experiments on TSP, CVRP and JSSP.

The paper then outlines related work followed by a detailed description of the problem setup and the algorithm. Section 4 describes the main contribution of the paper, which focuses on sampling method that enables sampling without replacement, but with improvement.  Here the paper discusses stochastic beam search and incremental sampling without improvement, as well as their Gumbledore sampling method. Next the paper provides details on theoretical policy improved and a description of the policy update method outlining how the sampling methods are integrated into the main algorithms. Section 5 describes the CO problems for which the proposed method shows the ability to solve CO problems at both small and large scales. The analysis also includes a section on sampling method performance. The paper then concludes with a discussion on limitations and a conclusion.

The main contributions of the paper focus on:
* Providing a new method for CO that trains a policy based on an iterative sampling approach that lies at the intersection of supervised learning and reinforcement learning.
* Showing scalability for the proposed method on different CO problems.

**Audience:**

Yes

**Claims And Evidence:**

Yes

**Requested Changes:**

Based on the weaknesses above, here are some requested changes:

**Important**:
* Provide evidence to strengthen the claim of sample efficiency for the proposed method.
* Add a discussion and traditional CO approaches and how they compare to the proposed method.
* More detailed table captions.

**Nice to Have**:
* Clarity on the transformer based architecture.

**Strengths And Weaknesses:**

**Strengths**:
* The paper proposed a new method for solving CO problems that lies at the intersection of supervised learning and reinforcement learning.
* The paper provides detailed discussion of methodological details.
* The experiments show scalability of the method and the importance of the proposed sampling method.

**Weakness:**
* I think the paper would benefit from showing some data on sample efficiency of the proposed method compared to other methods. This would help strengthen the claim of sample efficiency of the proposed method.
* The paper could benefit from a (potentially brief) broader discussion of traditional CO approaches, such as genetic algorithms, and position their proposed method compared to those.
* An ablation of a policy without a transformer based architecture could strengthen the claim for the importance of that choice.
* The paper could provide greater clarity in captions of Table 1 and Table 2; e.g., providing definitions for N and outlining the main takeaways.
* The paper could clarify how the sequence based data is processed by the proposed transformer architecture and define relevant hyperparameters for it.

---

> ### Author Response · Authors · 2024-05-16
> **Response to Reviewer WWT5 (Part 1/2)**
>
> Dear Reviewer,
> Thank you for taking the time to review our paper! In the following, we would like to respond to your requested changes and weaknesses (we took the liberty to group them). Changes in the updated manuscript are in blue font, and we indicate them in our response in **bold**).
>
> _**Weakness**: “I think the paper would benefit from showing some data on sample efficiency of the proposed method compared to other methods. This would help strengthen the claim of sample efficiency of the proposed method.”
> **With required change**: “Provide evidence to strengthen the claim of sample efficiency for the proposed method.”_
>
> **Response**: Thank you for this suggestion! While Section 5.3, “Sampling performance”, with Figure 3 shows the strength of the proposed Gumbeldore (GD) sampling in comparison to sampling with and without replacement on multiple checkpoints, we agree that it would be beneficial to see how it compares in the _entire training cycle_ from start to finish. Hence, we retrained all models with our proposed self-improvement algorithm, but we sampled sequences without replacement (WOR) (using stochastic beam search) instead of the advantage-based GD (keeping the computational budget identical). We chose to sample WOR, as it shares the diversity of stochastic beam search with GD and thus allows a fairer assessment if the advantage-based update of GD is beneficial (We note that GD reduces to WOR by setting the step size to $\sigma=0$). **The results are added as “SI WOR” in Tables 1 and 2 (and introduced in the main text at the top of page 9).** We can observe only a minimal difference for TSP (this is consistent with Figure 3, which shows that all sampling strategies yield good results for TSP). For the CVRP and JSSP, GD shows a remarkable improvement, supporting the claim of sampling efficiency of the proposed sampling method. Nevertheless, the results of WOR are still excellent (especially for the JSSP), so we believe that showing that our self-improvement strategy also works without the advantage-based update can foster adaptation of the method. **We add the trained model checkpoints for WOR to the existing ones in the supplementary material.**
> In the same spirit, **we furthermore added a curve in the plots of Figure 3**, where we apply GD with nucleus sampling completely switched off (i.e., GD with $p_{\text{min}} = 1$). This further underlines the benefits of the advantage-based update.
>
> Thank you for your suggestions, which have further strengthened the description of our method.
>
> _**Weakness**: “The paper could benefit from a (potentially brief) broader discussion of traditional CO approaches, such as genetic algorithms, and position their proposed method compared to those.”
> **With required change**: “Add a discussion and traditional CO approaches and how they compare to the proposed method.”_
>
> **Response**: A general comparison of neural CO methods with traditional heuristics and metaheuristics is insightful. For this purpose, we have **expanded the introduction to provide a bit more motivation for neural CO**, as well as the overall goal and why it diverts from traditional methods (as also requested by Reviewer 67nt).
> In addition, we decided to follow the same path as most NeurIPS/ICLR papers on neural CO (e.g., [4,5,6]) and **included the results of Google OR-Tools in Tables 1 and 2** (local search for routing problems and the strong CP-SAT for the JSSP).
> However, we think that a deeper comparison with traditional methods is out of scope for this paper, where we specifically aim to bridge the training problems attached to supervised/reinforcement learning in neural CO. We would like to point out that since neural CO has become a prominent research area in the last years, excellent review papers address this comparison and overall goals/strategies in neural CO, e.g., [1,2,3].
>
> We hope that you are satisfied with how we addressed this concern.

---

> > ### Comment · Reviewer_WWT5 · 2024-05-27
> > **Response to Response**
> >
> > Thank you for your thorough response and making edits to make the paper stronger. With regard to the discussion on traditional CO approaches, I think it would be good to mention that they exist and call out a couple of methods. This can be as little 1-3 sentences to provide additional context for paper readers.
> >
> > Other than that, I think that most of my feedback has been addressed.

---

> > > ### Author Response · Authors · 2024-05-27
> > > **Response to Reviewer WWT5**
> > >
> > > Dear Reviewer,
> > > Thank you for your additional feedback! We have incorporated your suggestion in the first paragraph of the introduction. Again, we sincerely appreciate your time and effort in reviewing our paper!

---

> ### Author Response · Authors · 2024-05-16
> **Response to Reviewer WWT5 (Part 2/2)**
>
> _**Weakness**: “An ablation of a policy without a transformer based architecture could strengthen the claim for the importance of that choice.”_
>
> **Response**: We believe you are referring to our proposed transformer-based architecture for the JSSP. We designed it this way due to the success of attention-based architectures in routing problems. Furthermore, our self-improvement method basically treats the policy training as a next-token prediction task. Decoder-only transformer models have shown to be very good at this, motivating the choice (for the routing problems, this is confirmed by the state-of-the-art reference BQ [3] and LEHD [4] architectures). **However, to make this more transparent in the manuscript, we made this next-token prediction training more precise in the description of the self-improvement training in Section 3.2 and explicitly Algorithm 1.**
>
> Regarding a policy without a transformer-based architecture, we would like to point to the neural CO method “SPN”, which we compare our approach to in Table 2. They train a graph neural network with a similar “self-labeling” strategy (there is a deeper comparison in Appendix E), exploiting the typical disjunctive graph formulation of the standard JSSP (their latent dimension is set to 128, our is set to 64). Since we consistently outperform SPN by a wide margin, this further indicates the suitability of the transformer-based choice.
>
> _**Weakness**: “The paper could provide greater clarity in captions of Table 1 and Table 2; e.g., providing definitions for N and outlining the main takeaways.”
> **With the required change**: “More detailed table captions.”_
>
> **Response**: Thanks for pointing this out. **We updated the table captions accordingly.**
>
> _**Weakness**: “The paper could clarify how the sequence based data is processed by the proposed transformer architecture and define relevant hyperparameters for it.
> **With required change (‘nice to have’)**: “Clarity on the transformer based architecture.”_
>
> **Response**: As noted in the response above, **we made the general training algorithm in Algorithm 1 and Section 3.2 more precise.** Regarding the individual architectures, we provide an overview of the BQ architecture for the routing problems in Appendix D.2.2 (TSP) and D.3.2 (CVRP). We note that the BQ [5] and LEHD [4] architectures are identical to the original work, so we only briefly overview the flow.
> For the JSSP architecture that we propose, we have assembled a very detailed description of the problem, schedule representation, and how it is processed in Appendix D.4. Together with the provided and documented source code, we believe it should be sufficient to reproduce or also reimplement the architecture. Please let us know if you are missing specific details.
>
> Thank you again for the valuable feedback, which helped refine the paper!
>
> ---
> **References:**
>
> [1] Bengio, Yoshua, Andrea Lodi, and Antoine Prouvost. "Machine learning for combinatorial optimization: a methodological tour d’horizon." European Journal of Operational Research 290.2 (2021): 405-421.
>
> [2] Liu, Shengcai, et al. "How good is neural combinatorial optimization? A systematic evaluation on the traveling salesman problem." IEEE Computational Intelligence Magazine 18.3 (2023): 14-28.
>
> [3] Mazyavkina, Nina, et al. "Reinforcement learning for combinatorial optimization: A survey." Computers & Operations Research 134 (2021): 105400.
>
> [4] Luo, Fu, et al. "Neural combinatorial optimization with heavy decoder: Toward large scale generalization." Advances in Neural Information Processing Systems 36 (2024).
>
> [5] Drakulic, Darko, et al. "BQ-NCO: Bisimulation Quotienting for Efficient Neural Combinatorial Optimization." Advances in Neural Information Processing Systems 36 (2024).
>
> [6] Zhang, Cong, et al. „Deep Reinforcement Learning Guided Improvement Heuristic for Job Shop Scheduling.“ The Twelfth International Conference on Learning Representations (2024)

---

### Review · Reviewer_SCRr · 2024-05-06

**Summary Of Contributions:**

The submission considers neural combinatorial optimization, where one uses neural-network-based methods for combinatorial optimization.

The proposed method is an improvement of classic behavior cloning (BC) methods. Specifically, the existing BC methods require lots of expert data, which can be hard to obtain, while the proposed method uses a novel sampling method to reduce the need of the expert data.

Experiment results are provided supporting the performance improvement of the proposed method.

**Audience:**

Yes

**Broader Impact Concerns:**

I do not have such concerns for this submission.

**Claims And Evidence:**

Yes

**Requested Changes:**

I find the submission well-written and self-contained. I do not have any specific suggestions on changes to the author.

**Strengths And Weaknesses:**

The submission is well written with clear motivations.

The proposed method is novel by using some existing techniques is a well-designed manner.

The performance improvement of the proposed method is thoroughly demonstrated empirically.

---

> ### Author Response · Authors · 2024-05-16
> **Response to Reviewer SCRr**
>
> Dear Reviewer,
> Thank you very much for your positive feedback and kind words about our paper. We are delighted to hear that you found our method well-designed and the manuscript well-written. We greatly appreciate your effort to review our paper!

---

### Review · Reviewer_67nt · 2024-05-13

**Summary Of Contributions:**

The paper is proposing a new approach for sampling sequences in the context of neural combinatorial optimization. The approach consist in a method for sampling a variety of likely sequence without duplicates in the set -- which duplicated seems likely to be obtained by purely random sampling according to the model. This lead to improvement to the use of self-improvement algorithm with random sampling. The approach is evaluated on several combinatorial optimization benchmarks, with good performances compared to the other baselines compared.

**Audience:**

Yes

**Broader Impact Concerns:**

I don't see specific broader impact concerns here, that more an algorithmic work aimed at improving combinatorial optimization.

**Claims And Evidence:**

Yes

**Requested Changes:**

Unfortunately, this paper is somehow outside my main expertise fields, I don't have specific technical changes requested to improve the paper. However, I find the paper to be somehow difficult to grasp at first for someone outside the neural combinatorial optimization field. I would like that the paper provide more background and help the reader to better understand the field, not assuming intimate knowledge with it. Also, the contribution make in the paper appears rather specific, for a niched algorithm, and as such of interest for a limited community. I would like the authors to develop more on that, and highlight well the contributions of the paper, For instance, it wasn't obvious to me whether the self-improvement neural combinatorial optimization presented in Algo. 1 is a proposal from some other paper or a novel proposal. The work is not very well contrasted with other work, for an outsider like me, the message is not very clear on the contributions and novelties.

**Strengths And Weaknesses:**

Strengths:
- Straightforward proposal to enhance self-improvement algorithm for combinatorial optimization.
- Clearly and carefully prepared papers.
- Results show good results on several combinatorial optimization benchmarks.

Weaknesses:
- Rather niche proposal for a specific algorithm. The interest for the approach might limited to a small community.
- Number of baselines used for comparison seem low, and I am wondering whether these baselines are the best to be used.

---

> ### Author Response · Authors · 2024-05-16
> **Response to Reviewer 67nt (Part 1/2)**
>
> Dear Reviewer,
>
> Thank you for reviewing our paper and for the constructive feedback! We highly appreciate the viewpoint outside the neural CO field that helps us make the paper more accessible. In the following, we would like to address your concerns. Changes in the updated manuscript are in blue font, and we indicate them in our response in **bold**).
>
> _“I would like that the paper provide more background and help the reader to better understand the field, not assuming intimate knowledge with it.“_
>
> **Response**: We agree that although neural CO has grown as a research field in recent years, our paper could be clearer on the background. Hence, **we extended the introduction to better introduce the field**. We want to summarize the main points and expand on some details for further explanation. We apologize if it becomes a bit lengthy:
>
> We briefly introduce CO, explain why neural CO is a promising approach, and discuss its ultimate goals (i.e., overcoming scalability issues and reducing the requirement for expert domain knowledge to design algorithms). We then introduce the constructive approach in neural CO, where a neural policy builds a solution incrementally. Due to its flexibility and natural modeling as a Markov decision process (suitable for reinforcement learning), the constructive approach is the primary approach in neural CO. How is this neural policy trained in the current literature? We describe the current methods used as either supervised learning (SL) or reinforcement learning (RL). In SL, the policy is trained on “expert solutions” (behavior cloning), which, e.g., for routing problems, come from traditional solvers (for example, for the Traveling Salesman Problem, from the Concorde solver, which builds on 50 years of research on linear programming and cutting plane algorithms [1]). Having these expert solutions is undoubtedly desirable. The big downside, however, is that obtaining solutions for SL can be computationally challenging or even impossible. This is where RL comes in, which uses the objective function of the problem as a reward to maximize and does not rely on pre-generated expert solutions. In general, however, the reward can only be given at the end of the episode, making the used algorithms susceptible to the sparse reward problem. The most widespread approach is using variants of the REINFORCE algorithm (e.g., with greedy rollouts of the current policy to baseline it). Actions are sampled until a complete solution is obtained, and the policy is optimized by gradient ascent over the final reward. The critical thing to note is that this requires accumulating gradients over complete trajectories, which can be computationally demanding (especially memory-wise).
>
> For routing problems, the omnipresent architecture used is the “Attention Model” (AM) [2], a transformer-based architecture that forms the backbone for most methods. Due to the computational cost mentioned, it performs the heavy lifting at the beginning on the “empty” problem instance (for example, all the cities/customers present in a routing problem) by encoding the instance with multiple attention layers, and then – during autoregressive decoding – only uses a single attention layer to compromise between performance and computational cost. This approach works remarkably well: POMO [3], for example, is still considered a state-of-the-art method for constructive neural CO. It builds on the AM and performs remarkably well on the training distribution of 20, 50, and 100 nodes. The problem, however, is that POMO (or other AM-based models) does not generalize well to larger instances. This is a known problem, and to complicate matters even further, it was observed recently (mainly by the two concurrent papers BQ [5] and LEHD [4] from last year’s NeurIPS) that the reason for this lies in the ‘lightweight’ decoder. BQ and LEHD substantially increase the decoder size (with BQ using a decoder-only architecture with nine transformer blocks, and LEHD single encoder block and six decoder blocks) and achieve new state-of-the-art results when generalizing to larger instances up to a thousand nodes. However, the larger architectures prohibit training with the RL methods above, which is why BQ and LEHD train their models with SL from expert data (apart from the specialized training strategy in the appendix of LEHD, which we compare in Appendix C.2).
>
> This “dilemma” (suitable architectures but relying on expert data) is where our paper comes in. In particular, the ‘self-improvement’ strategy described in Section 3.2 on the intersection of supervised and reinforcement learning is our novel proposal.
> We hope that the refined introduction makes this more evident.

---

> ### Author Response · Authors · 2024-05-16
> **Response to Reviewer 67nt (Part 2/2)**
>
> _“Also, the contribution make in the paper appears rather specific, for a niched algorithm, and as such of interest for a limited community. I would like the authors to develop more on that, and highlight well the contributions of the paper, For instance, it wasn't obvious to me whether the self-improvement neural combinatorial optimization presented in Algo. 1 is a proposal from some other paper or a novel proposal.”_
>
> **Response**: As outlined and motivated above, the self-improvement strategy is a novel proposal. Our self-improvement method basically treats the policy training as a next-token prediction task, **and we added more details in Section 3.** Furthermore, **we made the contributions at the end of the introduction more precise**, summarized as
>
> 1. The self-improvement training strategy (Section 3)
> 2. The sequence decoding method based on sampling without replacement with advantage-based updates. (Section 4)
> 3. Showing that we can train these larger architectures to a comparable performance without expert data.
> 4. Novel architecture for JSSP inspired by BQ and LEHD, outperforming state-of-the-art neural approaches.
>
> Hence, regarding your mentioned weakness:
>
> _“Rather niche proposal for a specific algorithm. The interest for the approach might limited to a small community.”_
>
> We agree that the proposal would be niche if contribution (ii) alone is taken (as a proposal for an improved sequence decoding method). However, with (i), we believe it’s of interest to a larger community within neural CO, as the self-improvement strategy is a general constructive method and proposes a diversion (and simplification) from the widespread RL methods used. To strengthen this, **we added results in Tables 1 and 2** for using the self-improvement strategy with sampling without replacement as a decoding method (we also refer to our response to Reviewer WWT5).
> We hope the abovementioned points become more evident in the updated manuscript.
>
> Finally, we would like to address the second weakness you mentioned
>
> _“Number of baselines used for comparison seem low, and I am wondering whether these baselines are the best to be used.“_
>
> We focus on the comparison with other neural CO methods. For the routing problems, the most important comparison partner to see if the self-improvement method works is the identical BQ architecture trained via SL with solutions from solvers (plus the LEHD architecture in the appendix). We also list two prominent baselines (the mentioned AM and POMO) for perspective and refer in the main text – as we use the same architecture – to the original BQ paper, where the authors make a sophisticated comparison with various other neural methods. However, we agree that referring to the paper might hurt our paper being self-contained, **so we added two additional neural baselines to Table 1 and provided explanations in the main text (Section 5.1, paragraph “Inference and baselines”) on how they relate to each other** (furthermore, in response to Reviewer WWT5, we added results from Google OR-Tools as a non-learning baseline).
>
> Regarding the Job Shop Scheduling Problem, the comparison partners L2D and ScheduleNet are state-of-the-art constructive methods. Notably, we further included L2S (a neural-guided improvement method) from this year’s ICLR, which already outperforms other neural methods by a large margin. Hence, we think that the chosen baselines are suitable for our claim. The results from SPN are included to compare to a method that uses a related training scheme based on sampling.
> (Similar to the routing problems, in response to Reviewer WWT5, we added results from CP-SAT within OR-Tools as a non-learning baseline)
>
> We hope our response and the changes made (especially to the introduction) address your concerns. We thank you again for the valuable feedback, which significantly helped to improve the manuscript!
>
> ---
>
> **References**:
>
> [1] https://iclr-blog-track.github.io/2022/03/25/deep-learning-for-routing-problems/
>
> [2] Kool, Wouter, Herke van Hoof, and Max Welling. "Attention, Learn to Solve Routing Problems!." International Conference on Learning Representations. 2018.
>
> [3] Kwon, Yeong-Dae, et al. "Pomo: Policy optimization with multiple optima for reinforcement learning." Advances in Neural Information Processing Systems 33 (2020): 21188-21198.
>
> [4] Luo, Fu, et al. "Neural combinatorial optimization with heavy decoder: Toward large scale generalization." Advances in Neural Information Processing Systems 36 (2024).
>
> [5] Drakulic, Darko, et al. "BQ-NCO: Bisimulation Quotienting for Efficient Neural Combinatorial Optimization." Advances in Neural Information Processing Systems 36 (2024).

---

> > ### Comment · Reviewer_67nt · 2024-05-28
> > **Thanks for the answers, I am satisfied with them**
> >
> > I have read carefully the answer to my reviews, as well as the other reviews and the corresponding answers. I have also looked at the updated paper. These are satisfying to me, it answers well my comments and requests.

---

> > > ### Author Response · Authors · 2024-05-29
> > > **Response to Reviewer 67nt**
> > >
> > > Dear Reviewer,
> > > Thank you again for your thorough review and your time; we are pleased that our revisions have satisfactorily addressed your comments!

---

### Author Response · Authors · 2024-05-16
**General response and summary of changes**

We sincerely thank all the reviewers and the action editor for their valuable feedback and support throughout the review process. We revised the paper according to your feedback and highlighted the changes in the manuscript in blue. We summarize them as follows:

- We have added a brief introduction to the motivation of neural combinatorial optimization at the beginning of the introduction (p. 1)
- We have expanded the introduction to make it easier to understand the problems with current forms of training and where exactly our paper begins (p. 2)
- At the end of the introduction, we have listed the contributions in a more explicit and concise manner (pp. 2-3)
- We provided more details on our proposed self-improvement training strategy (Algorithm 1) and clarified the contribution (pp. 4-5)
- We added experimental results for all problems of the self-improvement strategy when sampling sequences without replacement (no advantage-update/growing nucleus) to strengthen the claim of our method (Table 1 on p. 10, Table 2 on p. 12)
- We added results from Google’s OR-Tools for all problems (local search for routing, CP-SAT for JSSP) (Table 1 on p. 10, Table 2 on p. 12)
- We provide two additional baselines for the routing problems in Table 1 (p. 10) and relate them to each other on p. 9
- We give more detailed table captions (p. 10 and p. 12)
- We have added curves for the Gumbeldore sampling method without nucleus sampling to provide more evidence of its suitability (Figure 3, p. 13)
- We will add the model checkpoints for the additional experiments to the existing ones (currently not possible due to size constraints for the supplementary material)

---

### Comment · Action_Editor_z6TA · 2024-05-26
**[Reviewer Action neeed] Please respond to authors' responses**

Dear reviewers, please respond to the authors' responses - at least by acknowledging that you have read them. Use the opportunity to clarify any outstanding open questions. The author-reviewer discussion ends soon; after which we are aiming to quickly reach a decision.

---

### Decision · Action_Editor_z6TA · 2024-06-03

**Recommendation:** Accept as is

**Comment:**

**Summary:**
The paper addresses an interesting problem (combinatorial optimization, CO) with a modern and sophisticated approach. The idea of using a model's predictions for self-improvement has been proposed before, but its application to CO seems to be particularly fruitful. The general approach is to treat the solution to a CO problem as a sequence of actions and train a sequential predictor to assign high probability to (near-) optimal solution sequences. The innovation of the paper is to neither rely on expert data (imitate expert sequences) nor plain RL, but instead sample candidate solutions from the model, evaluate them, and retrain the model to increase the likelihood of the most successful samples. The paper presents a theoretically sound and efficient way to implement this idea with modern sequence models (transformers). This requires addressing two central problems:
 * The whole scheme depends on being able to efficiently sample "good" candidate solutions - the paper uses stochastic beam search (which has been proposed before in a different context) which allows for efficient (and parallelizable) batch-wise sampling *without replacement* from the model. This significantly improves the sample diversity without deteriorating sample quality, and thus allows for a structured exploration of a (partial) breadth-first search-tree.
 * The model needs to be bootstrapped on its own predictions - the paper presents a theoretically sound policy-improvement method for this and derives a practical version of it that can be combined well with stochastic beam search.
The method is evaluated on three classic CO problems: Travelling Salesman, Capacitated Vehicle Routing, and Job Shop Scheduling. Results highlight the efficiency and good performance of the method, which is further supported by comparisons against relevant baselines and gold-standard approaches.

**Verdict:**
The paper is very well written, and includes a great introduction to CO and the various standard approaches for a broad, non-expert audience. The innovations in the paper are well motivated and situated within the current literature, and the methodology is well-described. The method is theoretically sound, practical, uses modern techniques and sequence models, and works well empirically. While parts of the method have been discussed before in other context (bootstrapping for self-improvement, stochastic beam search, and parts of the policy improvement via Gumbel-top-k), the application to CO is novel and original, and putting together the individual parts of the method is non-trivial (and supported by ablations). All reviewers agree that the claims in the paper are supported by accurate, convincing and clear evidence, and that the work is interesting to TMLR's audience. I agree with their assessment and believe that the work is clearly ready and suitable for publication at TMLR.

**Certification:**
The paper is perhaps the highest-quality work that I have AE'd (7 papers) or reviewed (7 papers) for TMLR so far. The paper is very well written, and thanks to the great introduction to CO, and the literature review it is well accessible for a broad, non-expert audience. The method itself is original and well put together - the theoretical analysis nicely supports the method and does not bloat the manuscript with vacuous maths. Empirical results demonstrate that the method performs very well, and include great comparisons against SOTA and related methods, and important ablations. The paper is certainly interesting for a CO audience, but I believe that the method itself could also be highly relevant (at least highly inspiring) for using transformers in more general planning and control problems where expert data is not available and plain RL is prohibitively slow (the setting in the paper should straightforwardly translate to open-loop control and planning, and may be suitable to be included in feedback control and sequential decision-making). I therefore propose to consider the paper for a 'Featured' certification.

**Audience:**

Yes, the paper is highly relevant to a part of TMLR's audience.

**Claims And Evidence:**

Even before the authors' response (and revised manuscript) all reviewers agreed that all claims in the submission are supported by accurate, convincing and clear evidence. This was further strengthened by the author response and revised manuscript. All issues raised by the reviewers were sufficiently addressed, and no outstanding issues were mentioned by the reviewers.

Reading the final manuscript I agree with the reviewers' assessment that all claims are sufficiently supported by evidence.

---

> ### Author Response · Authors · 2024-06-07
> **Thank you**
>
> We are deeply grateful to all the reviewers and the AE for their insightful feedback and support. We greatly appreciate the effort invested in reviewing our work. Finally, we are deeply honored by the proposed certification!
>
> We have just uploaded the camera-ready version of the manuscript.
>
> Best regards,
> The authors